# Deep learning generalizes because the parameter-function map is biased towards simple functions

**Guillermo Valle Pérez**
University of Oxford
`guillermo.valle@dtc.ox.ac.uk`

**Chico Q. Camargo**
University of Oxford

**Ard A. Louis**
University of Oxford
`ard.louis@physics.ox.ac.uk`

## Abstract

Deep neural networks (DNNs) generalize remarkably well without explicit regularization even in the strongly over-parametrized regime where classical learning theory would instead predict that they would severely overfit. While many proposals for some kind of implicit regularization have been made to rationalise this success, there is no consensus for the fundamental reason why DNNs do not strongly overfit. In this paper, we provide a new explanation. By applying a very general probability-complexity bound recently derived from algorithmic information theory (AIT), we argue that the parameter-function map of many DNNs should be exponentially biased towards simple functions. We then provide clear evidence for this strong bias in a model DNN for Boolean functions, as well as in much larger fully conected and convolutional networks trained on CIFAR10 and MNIST. As the target functions in many real problems are expected to be highly structured, this intrinsic simplicity bias helps explain why deep networks generalize well on real world problems. This picture also facilitates a novel PAC-Bayes approach where the prior is taken over the DNN input-output function space, rather than the more conventional prior over parameter space. If we assume that the training algorithm samples parameters close to uniformly within the zero-error region then the PAC-Bayes theorem can be used to guarantee good expected generalization for target functions producing high-likelihood training sets. By exploiting recently discovered connections between DNNs and Gaussian processes to estimate the marginal likelihood, we produce relatively tight generalization PAC-Bayes error bounds which correlate well with the true error on realistic datasets such as MNIST and CIFAR10and for architectures including convolutional and fully connected networks.

## 1 Introduction

Deep learning is a machine learning paradigm based on very large, expressive and composable models, which most often require similarly large data sets to train. The name comes from the main component in the models: deep neural networks (DNNs) with many layers of representation. These models have been remarkably successful in domains ranging from image recognition and synthesis, to natural language processing, and reinforcement learning (Mnih et al. (2015); LeCun et al. (2015); Radford et al. (2016); Schmidhuber (2015)). There has been work on understanding the expressive power of certain classes of deep networks (Poggio et al. (2017)), their learning dynamics (Advani & Saxe (2017); Liao & Poggio (2017)), and generalization properties (Kawaguchi et al. (2017); Poggio et al. (2018); Neyshabur et al. (2017b)). However, a full theoretical understanding of many of these properties is still lacking.

DNNs are typically overparametrized, with many more parameters than training examples. Classical learning theory suggests that overparamterized models lead to overfitting, and so poorer generalization performance. By contrast, for deep learning there is good evidence that increasing the number of parameters leads to improved generalization (see e.g. Neyshabur et al. (2018)). For a typical supervised learning scenario, classical learning theory provides bounds on the generalization error $\epsilon(f)$ for target function $f$ that typically scale as the complexity of the hypothesis class $\mathcal{H}$. Complexity measures include simply the number of functions in $\mathcal{H}$, the VC dimension, and the Rademacher complexity (Shalev-Shwartz & Ben-David (2014)). Since neural networks are highly expressive, typical measures of $C(\mathcal{H})$ will be extremely large, leading to trivial bounds.

Many empirical schemes such as dropout (Srivastava et al. (2014)), weight decay (Krogh & Hertz (1992)), early stopping (Morgan & Bourlard (1990)), have been proposed as sources of regularization that effectively lower $C(\mathcal{H})$. However, in an important recent paper (Zhang et al. (2017a)), it was explicitly demonstrated that these regularization methods are not necessary to obtain good generalization. Moreover, by randomly labelling images in the well known CIFAR10 data set (Krizhevsky & Hinton (2009)), these authors showed that DNNs are sufficiently expressive to memorize a data set in not much more time than it takes to train on uncorrupted CIFAR10 data. By showing that it is relatively easy to train DNNs to find functions $f$ that do not generalize at all, this work sharpened the question as to why DNNs generalize so well when presented with uncorrupted training data. This study stimulated much recent work, see e.g. (Kawaguchi et al. (2017); Arora et al. (2018); Morcos et al. (2018); Neyshabur et al. (2017b); Dziugaite & Roy (2017; 2018); Neyshabur et al. (2017a; 2018)), but there is no consensus as to why DNNs generalize so well.

Because DNNs have so many parameters, minimizing the loss function $L$ to find a minimal training set error is a challenging numerical problem. The most popular methods for performing such *optimization* rely on some version of stochastic gradient descent (SGD). In addition, many authors have also argued that SGD may exploit certain features of the loss-function to find solutions that generalize particularly well (Soudry et al. (2017); Zhang et al. (2017b; 2018)) However, while SGD is typically superior to other standard minimization methods in terms of optimization performance, there is no consensus in the field on how much of the remarkable generalization performance of DNNs is linked to SGD (Krueger et al. (2017)). In fact DNNs generalize well when other optimization methods are used (from variants of SGD, like Adam (Kingma & Ba (2014)), to gradient-free methods (Such et al. (2018))). For example, in recent papers (Wu et al. (2017); Zhang et al. (2018); Keskar et al. (2016) simple gradient descent (GD) was shown to lead to differences in generalization with SGD of at most a few percent. Of course in practical applications such small improvements in generalization performance can be important. However, the question we want to address in this paper is the broader one of **Why do DNNs generalize at all, given that they are so highly expressive and overparameterized?**. While SGD is important for optimization, and may aid generalization, it does not appear to be the fundamental source of generalization in DNNs.

Another longstanding family of arguments focuses on the local curvature of a stationary point of the loss function, typically quantified in terms of products of eigenvalues of the local Hessian matrix. Flatter stationary points (often simply called minima) are associated with better generalization performance (Hochreiter & Schmidhuber (1997); Hinton & van Camp (1993)). Part of the intuition is that flatter minima are associated with simpler functions (Hochreiter & Schmidhuber (1997); Wu et al. (2017)), which should generalize better. Recent work (Dinh et al. (2017)) has pointed out that flat minima can be transformed to sharp minima under suitable re-scaling of parameters, so care must be taken in how flatness is defined. In an important recent paper (Wu et al. (2017)) an attack data set was used to vary the generalization performance of a standard DNN from a few percent error to nearly $100\%$ error. This performance correlates closely with a robust measure of the flatness of the minima (see also Zhang et al. (2018) for a similar correlation over a much smaller range, but with evidence that SGD leads to slightly flatter minima than simple GD does). The authors also conjectured that this large difference in local flatness would be reflected in large differences between the volume $V_{\text{good}}$ of the basin of attraction for solutions that generalize well and $V_{\text{bad}}$ for solutions that generalize badly. If these volumes differ sufficiently, then this may help explain why SGD and other methods such as GD converge to good solutions; these are simply much easier to find than bad ones. Although this line of argument provides a tantalizing suggestion for why DNNs generalize well in spite of being heavily overparameterized, it still begs the fundamental question of **Why do solutions vary so much in flatness or associated properties such as basin volume?**

In this paper we build on recent applications of algorithmic information theory (AIT)(Dingle et al. (2018)) to suggest that the large observed differences in flatness observed by (Wu et al. (2017)) can be correlated with measures of descriptional complexity. We then apply a connection between Gaussian processes and DNNS to empirically demonstrate for several different standard architectures that the probability of obtaining a function $f$ in DNNs upon a random choice of parameters varies over many orders of magnitude. This bias allows us to apply a classical result from PAC-Bayes theory to help explain why DNNs generalize so well.

## 1.1 Main contributions

Our main contributions are:

- We argue that the parameter-function map provides a fruitful lens through which to analyze the generalization performance of DNNs.

- We apply recent arguments from AIT to show that, if the parameter-function map is highly biased, then high probability functions will have low descriptional complexity.

- We show empirically that the parameter-funtion map of DNNs is extremely biased towards simple functions, and therefore the prior over functions is expected to be extremely biased too. We claim that this intrinsic bias towards simpler functions is the fundamental source of regularization that allows DNNs to generalize.

- We approximate the prior over functions using Gaussian processes, and present evidence that Gaussian processes reproduce DNN marginal likelihoods remarkably well even for finite width networks.

- Using the Gaussian process approximation of the prior over functions, we compute PAC-Bayes expected generalization error bounds for a variety of common DNN architectures and datasets. We show that this shift from the more commonly used priors over parameters to one over functions allows us to obtain relatively tight bounds which follow the behaviour of the real generalization error.

## 2 The parameter-function map

**Definition 1.** *(Parameter-function map) For a parametrized supervised learning model, let the input space be $\mathcal{X}$ and the output space be $\mathcal{Y}$. The space of functions that the model can express is then $\mathcal{F} \subseteq \mathcal{Y}^{|\mathcal{X}|}$. If the model has $p$ real valued parameters, taking values within a set $\Theta \subseteq \mathbb{R}^p$, the parameter-function map, $\mathcal{M}$, is defined as:*

$$\mathcal{M} : \Theta \to \mathcal{F}$$
$$\theta \mapsto f_\theta$$

*where $f_\theta$ is the function implemented by the model with choice of parameter vector $\theta$.*

This map is of interest when using an algorithm that searches in parameter space, such as stochastic gradient descent, as it determines how the behaviour of the algorithm in parameter space maps to its behavior in function space. The latter is what determines properties of interest like generalization.

## 3 Algorithmic information theory and simplicty-bias in the parameter-function map

One of the main sources of inspiration for our current work comes from a recent paper (Dingle et al. (2018)) which derives an upper bound for the probability $P(x)$ that an output $x \in O$ of an input-output map $g : I \to O$ obtains upon random sampling[1] of inputs $I$

$$P(x) \leq 2^{-a\tilde{K}(x)+b} \tag{1}$$

---

[1]We typically assume uniform sampling of inputs, but other distributions with simple descriptional complexity would also work

where $\tilde{K}(x)$ is an approximation to the (uncomputable) Kolmogorov complexity of $x$, and $a$ and $b$ are scalar parameters which depend on the map $g$ but not on $x$. This bound can be derived for inputs $x$ and maps $g$ that satisfy a simplicity criterion $K(g) + K(n) \ll K(x) + \mathcal{O}(1)$, where $n$ is a measure of the size of the inputs space (e.g. if $|I| = 2^n$). A few other conditions also need to be fulfilled in order for equation (1) to hold, including that there is redundancy in the map, e.g. that multiple inputs map to the same output $x$ so that $P(x)$ can vary (for more details see Appendix E)). Statistical lower bounds can also be derived for $P(x)$ which show that, upon random sampling of inputs, outputs $x$ will typically be close (on a log scale) to the upper bound.

If there is a significant linear variation in $K(x)$, then the bound (1) will vary over many orders of magnitude. While these arguments (Dingle et al. (2018)) do not prove that maps are biased, they do say that if there is bias, it will follow Eq. (1). It was further shown empirically that Eq. (1) predicts the behavior of maps ranging from the RNA sequence to secondary structure map to a map for option pricing in financial mathematics very well.

It is not hard to see that the DNN parameter-function map $\mathcal{M}$ is simple in the sense described above, and also fulfills the other necessary conditions for simplicity bias (Appendix E). The success of Eq. (1) for other maps suggests that the parameter-function map of many different kinds of DNN should also exhibit simplicity bias.

## 4 SIMPLICITY BIAS IN A DNN IMPLEMENTING BOOLEAN FUNCTIONS

In order to explore the properties of the parameter-function map, we consider *random neural networks*. We put a probability distribution over the space of parameters $\Theta$, and are interested in the distribution over functions induced by this distribution via the parameter-function map of a given neural network. To estimate the probability of different functions, we take a large sample of parameters and simply count the number of samples producing individual functions (*empirical frequency*).

This procedure is easiest for a discrete space of functions, and a small enough function space so that the probabilities of obtaining functions more than once is not negligible. We achieve this by using a neural network with 7 Boolean inputs, two hidden layers of 40 ReLU neurons each, and a single Boolean output. This means that the input space is $\mathcal{X} = \{0,1\}^7$ and the space of functions is $\mathcal{F} \subseteq \{0,1\}^{2^7}$ (we checked that this neural network can in fact express almost all functions in $\{0,1\}^{2^7}$). For the distributions on parameter space we used Gaussian distributions or uniform within a hypercube, with several variances[2]

The resulting probabilities $P(f)$ can be seen in Figure 1a, where we plot the (normalized) empirical frequencies versus the rank, which exhibits a range of probabilities spanning as many orders of magnitude as the finite sample size allows. Using different distributions over parameters has a very small effect on the overall curve.

We show in Figure 1a that the rank plot of $P(f)$ can be accurately fit with a normalized Zipf law $P(r) = (\ln(N_O)r)^{-1}$, where $r$ is the rank, and $N_O = 2^{128}$. It is remarkable that this fits the rank plot so well without any adjustable parameters, suggesting that Zipf behaviour is seen over the whole rank plot, leading to an estimated range of 39 orders of magnitude in $P(f)$. Also, once the DNN is sufficiently expressive so that all $N_O$ functions can be found, this arguments suggests that $P(f)$ will not vary much with an increase in parameters. Independently, Eq. (1) also suggests that to first order $P(f)$ is independent of the number of parameters since if follows from $K(f)$, and the map only enters through $a$ and $b$, which are constrained (Dingle et al. (2018)).

The bound of equation (1) predicts that high probability functions should be of low descriptional complexity. In Figure 10, we show that we indeed find this simplicity-bias phenomenon, with all high-probability functions having low Lempel-Ziv (LZ) complexity (defined in Appendix F.1). The literature on complexity measures is vast. Here we simply note that there is nothing fundamental about LZ. Other approximate complexity measures that capture essential aspects of Kolmogorov complexity also show similar correlations. In Appendix F.4, we demonstrate that probability also correlates with the size of the smallest Boolean expression expressing that function, as well as with two measures of the sensitivity of the output to changes in the input. In Figure 9, we also compare the different measures. We can see that they all correlate, but also show differences in their ability

---

[2]for Figures 1b,11 we used a uniform distribution with variance of $1/\sqrt{n}$ with $n$ the input size to the layer

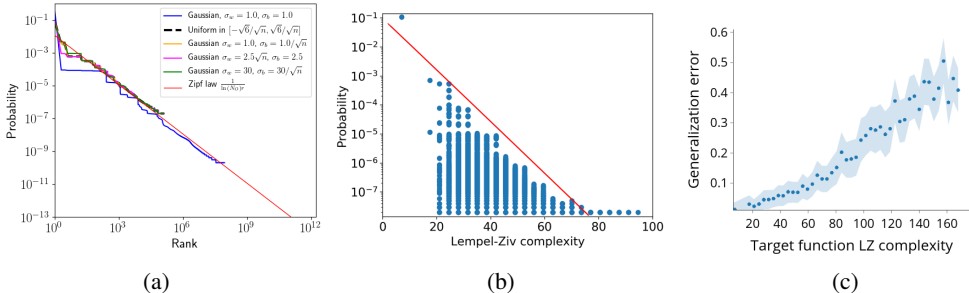

Figure 1: (a) Probability versus rank of each of the functions (ranked by probability) from a sample of $10^{1}0$ (blue) or $10^7$ (others) parameters. The labels are different parameter distributions. (b) Probability versus Lempel-Ziv complexity. Probabilities are estimated from a sample of $10^8$ parameters. Points with a frequency of $10^{-8}$ are removed for clarity because these suffer from finite-size effects (see Appendix G). The red line is the simplicity bias bound of Eq.(1) (d) Generalization error increases with the Lempel-Ziv complexity of different target functions, when training the network with advSGD – see Appendix A.

to recognize regularity. It is still an open question, which are the complexity measures which better capture real-world regularity.

There are many reasons to believe that real-world functions are simple or have some structure (Lin et al. (2017); Schmidhuber (1997)) and will therefore have low descriptional complexity. Putting this together with the above results means we expect a DNN that exhibits simplicity bias to generalize well for real-world datasets and functions. On the other hand, as can also be seen in Figure 1c, our network does not generalize well for complex (random) functions. By simple counting arguments, the number of high complexity functions is exponentially larger than the number of low complexity functions. Nevertheless, such functions may be less common in real-world applications.

Although here we have only shown that high-probability functions have low complexity for a relatively small DNN, the generality of the AIT arguments from Dingle et al. (2018) suggests that an exponential probability-complexity bias should hold for larger neural networks as well. To test this for larger networks, we restricted the input to a random sample of 1000 images from CIFAR10, and then sampled networks parameters for a 4 layer CNN, to sample labelings of these inputs. We then computed the probability of these networks, together with the critical sample ratio (a complexity measured defined in Appendix F.1) of the network on these inputs. As can be seen in Figure 2a they correlate remarkably well, suggesting that simplicity bias is also found for more realistic DNNs.

We next turn to a complementary approach from learning theory to explore the link between bias and generalization.

## 5 PAC-Bayes generalization error bounds

Following the failure of worst-case generalization bounds for deep learning, algorithm and data-dependent bounds have been recently investigated. The main three approaches have been based on algorithmic stability (Hardt et al. (2016)), margin theory (Neyshabur et al. (2015); Keskar et al. (2016); Neyshabur et al. (2017b;a); Bartlett et al. (2017); Golowich et al. (2018); Arora et al. (2018); Neyshabur et al. (2018)), and PAC-Bayes theory (Dziugaite & Roy (2017; 2018); Neyshabur et al. (2017a;b)).

Here we build on the classic PAC-Bayes theorem by McAllester (1999a) which bounds the expected generalization error when picking functions according to some distribution $Q$ (which can depend on the training set), given a (training-set independent) prior $P$ over functions. Langford & Seeger (2001) tightened the bound into the form shown in Theorem 1.

**Theorem 1.** *(PAC-Bayes theorem (Langford & Seeger (2001)) For any distribution $P$ on any concept space and any distribution $\mathcal{D}$ on a space of instances we have, for $0 < \delta \leq 1$, that with*

*probability at least $1 - \delta$ over the choice of sample $S$ of $m$ instances, all distributions $Q$ over the concept space satisfy the following:*

$$\hat{\epsilon}(Q) \ln \frac{\hat{\epsilon}(Q)}{\epsilon(Q)} + (1 - \hat{\epsilon}(Q)) \ln \frac{1 - \hat{\epsilon}(Q)}{1 - \epsilon(Q)} \leq \frac{\mathrm{KL}(Q\|P) + \ln\left(\frac{2m}{\delta}\right)}{m - 1} \tag{2}$$

*where $\epsilon(Q) = \sum_c Q(c)\epsilon(c)$, and $\hat{\epsilon}(Q) = \sum_c Q(c)\hat{\epsilon}(c)$. Here, $\epsilon(c)$ is the generalization error (probability of the concept $c$ disagreeing with the target concept, when sampling inputs according to $\mathcal{D}$), and $\hat{\epsilon}(c)$ is the empirical error (fraction of samples in $S$ where $c$ disagrees with the target concept).*

In the *realizable* case (where zero training error is achievable for any training sample of size $m$), we can consider an algorithm that achieves zero training error and samples functions with a weight proportional the prior, namely $Q(c) = Q^*(c) = \frac{P(c)}{\sum_{c \in U} P(c)}$, where $U$ is the set of concepts consistent with the training set. This is just the posterior distribution, when the prior is $P(c)$ and likelihood equals 1 if $c \in U$ and 0 otherwise. It also is the $Q$ that minimizes the general PAC-Bayes bound 2 (McAllester (1999a)). In this case, the KL divergence in the bound simplifies to the marginal likelihood (Bayesian evidence) of the data[3], and the right hand side becomes an invertible function of the error. This is shown in Corollary 1, which is just a tighter version of the original bound by McAllester (1998) (Theorem 1) for Bayesian binary classifiers. In practice, modern DNNs are often in the realizable case, as they are often trained to reach 100% training accuracy.

**Corollary 1.** *(Realizable PAC-Bayes theorem (for Bayesian classifier)) Under the same setting as in Theorem 1, with the extra assumption that $\mathcal{D}$ is realizable, we have:*

$$-\ln(1 - \epsilon(Q^*)) \leq \frac{\ln \frac{1}{P(U)} + \ln\left(\frac{2m}{\delta}\right)}{m - 1}$$

*where $Q^*(c) = \frac{P(c)}{\sum_{c \in U} P(c)}$, $U$ is the set of concepts in $\mathcal{H}$ consistent with the sample $S$, and where $P(U) = \sum_{c \in U} P(c)$*

Here we interpret $\epsilon(Q)$ as the expected value of the generalization error of the classifier obtained after running a stochastic algorithm (such as SGD), where the expectation is over runs. In order to apply the PAC-Bayes corollary(which assumes sampling according to $Q^*$), we make the following (informal) assumption:

*Stochastic gradient descent samples the zero-error region close to uniformly.*

Given some distribution over parameters $\tilde{P}(\theta)$, the distribution over functions $P(c)$ is determined by the parameter-function map as $P(c) = \tilde{P}(\mathcal{M}^{-1}(c))$. If the parameter distribution is not too far from uniform, then $P(c)$ should be heavily biased as in Figure 1a. In Section 7, we will discuss and show further evidence for the validity of this assumption on the training algorithm. One way to understand the bias observed in Fig 1a is that the volumes of regions of parameter space producing functions vary exponentially. This is likely to have a very significant effect on which functions SGD finds. Thus, even if the parameter distributions used here do not capture the exact behavior of SGD, the bias will probably still play an important role.

Our measured large variation in $P(f)$ should correlate with a large variation in the basin volume $V$ that Wu et al. (2017) used to explain why they obtained similar results using GD and SGD for their DNNs trained on CIFAR10.

Because the region of parameter space with zero-error may be unbounded, we will use, unless stated otherwise, a Gaussian distribution with a sufficiently large variance[4]. We discuss further the effect of the choice of variance in Appendix C.

---

[3]This can be obtained, for instance, by noticing that the KL divergence between $Q$ and $P$ equals the evidence lower bound (ELBO) plus the log likelihood. As $Q^*$ is the true posterior, the bound becomes an equality, and in our case the log likelihood is zero.

[4]Note that in high dimensions a Gaussian distribution is very similar to a uniform distribution over a sphere.

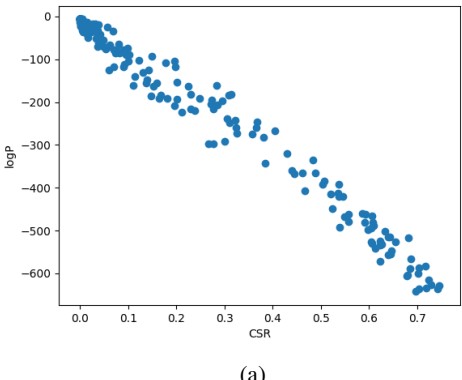
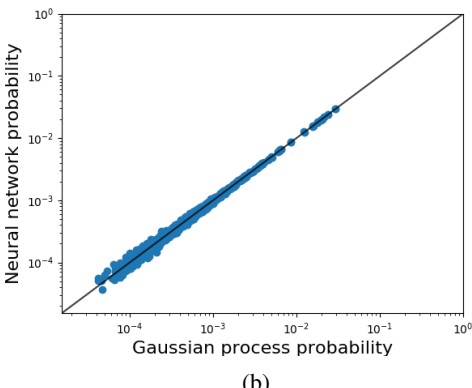

(a)                                                    (b)

Figure 2: (a) Probability (using GP approximation) versus critical sample ratio (CSR) of labellings of 1000 random CIFAR10 inputs, produced by 250 random samples of parameters. The network is a 4 layer CNN. (b) Comparing the empirical frequency of different labelings for a sample of $m$ MNIST images, obtained from randomly sampling parameters from a neural neural network, versus that obtained by sampling from the corresponding GP. The network has 2 fully connected hidden layers of 784 ReLU neurons each. $\sigma_w = \sigma_b = 1.0$. Sample size is $10^7$, and only points obtained in both samples are displayed. These figures also demonstrate significant (simplicity) bias for $P(f)$.

In order to use the PAC-Bayes approach, we need a method to calculate $P(U)$ for large systems, a problem we now turn to.

## 5.1 GAUSSIAN PROCESS APPROXIMATION TO THE PRIOR OVER FUNCTIONS

In recent work (Lee et al. (2017); Matthews et al. (2018); Garriga-Alonso et al. (2018); Novak et al. (2018)), it was shown that infinitely-wide neural networks (including convolutional and residual networks) are equivalent to Gaussian processes. This means that if the parameters are distributed i.i.d. (for instance with a Gaussian with diagonal covariance), then the (real-valued) outputs of the neural network, corresponding to any finite set of inputs, are jointly distributed with a Gaussian distribution. More precisely, assume the i.i.d. distribution over parameters is $\tilde{P}$ with zero mean, then for a set of $n$ inputs $(x_1, ..., x_n)$,

$$P_{\theta \sim \tilde{P}}\left(f_\theta(x_1) = \tilde{y}_1, ..., f_\theta(x_n) = \tilde{y}_n\right) \propto \exp\left(-\frac{1}{2}\tilde{\mathbf{y}}^T \mathbf{K}^{-1} \tilde{\mathbf{y}}\right), \tag{3}$$

where $\tilde{\mathbf{y}} = (\tilde{y}_1, ..., \tilde{y}_n)$. The entries of the covariance matrix $\mathbf{K}$ are given by the kernel function $k$ as $K_{ij} = k(x_i, x_j)$. The kernel function depends on the choice of architecture, and properties of $\tilde{P}$, in particular the weight variance $\sigma_w^2/n$ (where $n$ is the size of the input to the layer) and the bias variance $\sigma_b^2$. The kernel for fully connected ReLU networks has a well known analytical form known as the arccosine kernel (Cho & Saul (2009)), while for convolutional and residual networks it can be efficiently computed[5].

The main quantity in the PAC-Bayes theorem, $P(U)$, is precisely the probability of a given set of output labels for the set of instances in the training set, also known as *marginal likelihood*, a connection explored in recent work (Smith & Le (2017); Germain et al. (2016)). For binary classification, these labels are binary, and are related to the real-valued outputs of the network via a nonlinear function such as a step functionwhich we denote $\sigma$. Then, for a training set $U = \{(x_1, y_1), ..., (x_m, y_m)\}$, $P(U) = P_{\theta \sim \tilde{P}}(\sigma(f_\theta(x_1)) = y_1, ..., \sigma(f_\theta(x_m)) = y_m)$.

This distribution no longer has a Gaussian form because of the output nonlinearity $\sigma$. We will discuss how to circumvent this. But first, we explore the more fundamental issue of neural networks not

---

[5]We use the code from Garriga-Alonso et al. (2018) to compute the kernel for convolutional networks

being infinitely-wide in practice[6]. To test whether the equation above provides a good approximation for $P(U)$ for common neural network architectures, we sampled **y** (labellings for a particular set of inputs) from a fully connected neural network, and the corresponding Gaussian process, and compared the empirical frequencies of each function. We can obtain good estimates of $P(U)$ in this direct way, for very small sets of inputs (here we use $m = 10$ random MNIST images). The results are plotted in Figure 2, showing that the agreement between the neural network probabilities and the Gaussian probabilities is extremely good, even this far from the infinite width limit (and for input sets of this size).

In order to calculate $P(U)$ using the GPs, we use the expectation-propagation (EP) approximation, implemented in GPy (since 2012), which is more accurate than the Laplacian approximation (see Rasmussen (2004) for a description and comparison of the algorithms). To see how good these approximations are, we compared them with the empirical frequencies obtained by directly sampling the neural network. The results are in Figure 5 in the Appendix B. We find that the both the EP and Laplacian approximations correlate with the the empirical neural network likelihoods. In larger sets of inputs (1000), we also found that the relative difference between the log-likelihoods given by the two approximations was less than about 10%.

## 6 EXPERIMENTAL RESULTS

We tested the expected generalization error bounds described in the previous section in a variety of networks trained on binarized[7] versions of MNIST (LeCun et al. (1998)), fashion-MNIST (Xiao et al. (2017)), and CIFAR10 (Krizhevsky & Hinton (2009)). Zhang et al. (2017a) found that the generalization error increased continuously as the labels in CIFAR10 where randomized with an increasing probability. In Figure 3, we replicate these results for three datasets, and show that our bounds correctly predict the increase in generalization error. Furthermore, the bounds show that, for low corruption, MNIST and fashion-MNIST are similarly hard (although fashion-MIST is slightly harder), and CIFAR10 is considerably harder. This mirrors what is obtained from the true generalization errors. Also note that the bounds for MNIST and fashion-MNIST with little corruption are significantly below 0.5 (random guessing). For experimental details see Appendix A.

In the inset of Figure 3, we show that $P(U)$ decreases over many orders of magnitude with increasing label corruption, which is a proxy for complexity.

In Table 1, we list the mean generalisation error and the bounds for the three datasets (at 0 label corruption), demonstrating that the PAC-Bayes bound closely follows the same trends.

| Network | MNIST | | fashion-MNIST | | CIFAR | |
|---------|-------|-------|---------------|-------|-------|-------|
|         | Error | Bound | Error | Bound | Error | Bound |
| CNN | 0.023 | 0.134 | 0.071 | 0.175 | 0.320 | 0.485 |
| FC  | 0.031 | 0.169 | 0.070 | 0.188 | 0.341 | 0.518 |

Table 1: Mean generalization errors and PAC-Bayes bounds for the convolutional and fully connected network for 0 label corruption, for a sample of 10000 from different datasets. The networks are the same as in Figure 3 (4 layer CNN and 1 layer FC)

## 7 SGD VERSUS BAYESIAN SAMPLING

In this section we test the assumption that SGD samples parameters close to uniformly within the zero-error region. In the literature, Bayesian sampling of the parameters of a neural network has been argued to produce generalization performance similar to the same network trained with SGD. The evidence is based on comparing SGD-trained network with a Gaussian process approximation (Lee et al. (2017)), as well as showing that this approximation is similar to Bayesian sampling via MCMC methods (Matthews et al. (2018)).

---

[6]Note that the Gaussian approximation has been previously used to study finite neural networks as a mean-field approximation (Schoenholz et al. (2017a)).

[7]We label an image as 0 if it belongs to one of the first five classes and as 1 otherwise

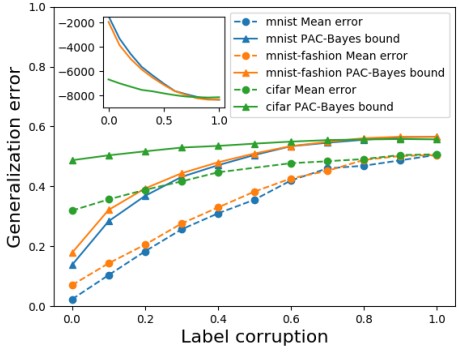
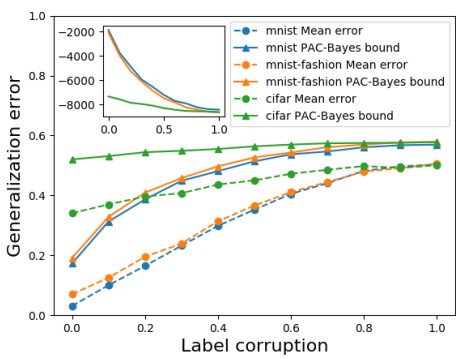

(a) for a 4 hidden layers convolutional network    (b) for a 1 hidden layer fully connected network

Figure 3: Mean generalization error and corresponding PAC-Bayes bound versus percentage of label corruption, for three datasets and a training set of size $10000$. Training set error is $0$ in all experiments. Note that the bounds follow the same trends as the true generalization errors. The empirical errors are averaged over $8$ initializations. The Gaussian process parameters were $\sigma_w = 1.0$, $\sigma_b = 1.0$ for the CNN and $\sigma_w = 10.0$, $\sigma_b = 10.0$ for the FC. **Insets** show the marginal likelihood of the data as computed by the Gaussian process approximation (in natural log scale), versus the label corruption.

We performed experiments showing direct evidence that the probability with which two variants of SGD find functions is close to the probability of obtaining the function by uniform sampling of parameters in the zero-error region. Due to computational limitations, we consider the neural network from Section 4. We are interested in the probability of finding individual functions consistent with the training set, by two methods:(1 Training the neural network with variants of SGD[8]; in particular, advSGD and Adam (described in Appendix A) (2 Bayesian inference using the Gaussian process corresponding to the neural network architecture. This approximates the behavior of sampling parameters close to uniformly in the zero-error region (i.i.d. Gaussian prior to be precise).

We estimated the probability of finding individual functions, averaged over training sets, for these two methods (see Appendix D for the details), when learning a target Boolean function of LZ complexity$84.0$. In Figures 4 and 8, we plot this average probability, for an SGD-like algorithm, and for the approximate Bayesian inference. We find that there is close agreement (specially taking into account that the EP approximation we use appears to overestimate probabilities, see Appendix B), although with some scatter (the source of which is hard to discern, given that the SGD probabilities have sampling error).

These results are promising evidence that SGD may behave similarly to uniform sampling of parameters (within zero-error region). However, this is still a question that needs much further work. We discuss in Appendix C some potential evidence for SGD sometimes diverging from Bayesian parameter sampling.

## 8    CONCLUSION AND FUTURE WORK

In this paper, we present an argument that we think offers a first-order explanation of generalization in highly overparameterized DNNs. First, PAC-Bayes shows how priors which are sufficiently biased towards the true distribution can result in generalization in highly expressive models, e.g. even if there are many more parameters than data points. Second, the huge bias towards simple functions in the parameter-function map strongly suggests that neural networks have a similarly biased prior. The number of parameters in a fully expressive DNN does not strongly affect the bias. Third, since real-world problems tend to be far from random, using these same complexity measures, we expect the prior to be biased towards the right class of solutions for real-world datasets and problems.

---

[8]These methods were chosen because other methods we tried, including plain SGD, didn't converge to zero error in this task

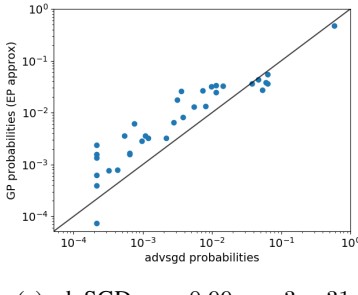

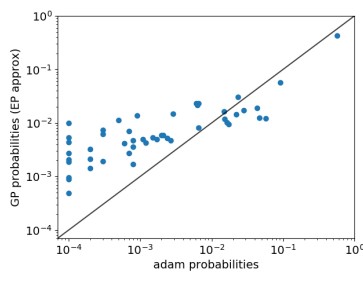

(a) advSGD. $\rho = 0.99, p = 3\mathrm{e}{-31}$         (b) Adam. $\rho = 0.99, p = 8\mathrm{e}{-42}$

Figure 4: Average probability of finding a function for a variant of SGD, versus average probability of finding a function when using the Gaussian process approximation. This is done for a randomly chosen, but fixed, target Boolean function of Lempel-Ziv complexity $84.0$. See Appendix D for details. The Gaussian process parameters are $\sigma_w = 10.0$, and $\sigma_b = 10.0$. For advSGD, we have removed functions which only appared once in the whole sample, to avoid finite-size effects. In the captions, $\rho$ refers to the 2-tailed Pearson correlation coefficient, and $p$ to its corresponding p value.

To demonstrate the bias in the parameter-function map, we used both direct sampling for a small network and an equivalence with Gaussian processes for larger networks. We also used arguments from AIT to show that functions $f$ that obtain with higher $P(f)$ are likely to be simple. However, a more complete understanding of this bias is still called for.

We also demonstrated how to make this approach quantitative, approximating neural networks as Gaussian processes to calculate PAC-Bayesian bounds on the generalization error.

It should be noted that our approach is not yet able to explain the effects that different tricks used in practice have on generalization. However, most of these improvements tend to be of the order of a few percent in the accuracy. The aim of this paper is to explain the bulk of the generalization, which classical learning theory would predict to be poor in this highly overparametrized regime. It is still an open question whether our approach can be extended to explain some of the tricks used in practice, as well as to methods other than neural networks (Belkin et al. (2018)) that may also have simple parameter-function maps. To stimulate work in improving our bounds, we summarize here the main potential sources of error for our bounds:

1. The probability that the training algorithm (like SGD) finds a particular function in the zero-error region can be approximated by the probability that the function obtains upon i.i.d. sampling of parameters.

2. Gaussian processes model neural networks with i.i.d.-sampled parameters well even for finite widths.

3. Expectation-propagation gives a good approximation of the Gaussian process marginal likelihood.

4. PAC-Bayes offers tight bounds given the correct marginal likelihood $P(U)$.

We have shown evidence that number 2 is a very good approximation, and that numbers 1 and 3 are reasonably good. In addition, the fact that our bounds are able to correctly predict the behavior of the true error, offers evidence for the set of approximations as a whole, although further work in testing their validity is needed, specially that of number 1. Nevertheless, we think that the good agreement of our bounds constitutes good evidence for the approach we describe in the paper as well as for the claim that bias in the parameter-function map is the main reason for generalization. We think that further work in understanding these assumptions can sharpen the results obtained here significantly.

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

## A    BASIC EXPERIMENTAL DETAILS

In the main experiments of the paper we used two classes of architectures. Here we describe them in more detail.

- Fully connected networks (FCs), with varying number of layers. The size of the hidden layers was the same as the input dimension, and the nonlinearity was ReLU. The last layer was a single Softmax neuron. We used default Keras settings for initialization (Glorot uniform).
- Convolutional neural networks (CNNs), with varying number of layers. The number of filters was 200, and the nonlinearity was ReLU. The last layer was a fully connected single Softmax neuron. The filter sizes alternated between $(2, 2)$ and $(5, 5)$, and the padding between SAME and VALID, the strides were $1$ (same default settings as in the code for Garriga-Alonso et al. (2018)). We used default Keras settings for initialization (Glorot uniform).

In all experiments in Section 6 we trained with SGD with a learning rate of $0.01$, and early stopping when the accuracy on the whole training set reaches $100\%$.

In the experiments where we learn Boolean functions with the smaller neural network with 7 Boolean inputs and one Boolean output (results in Figure 1c), we use a variation of SGD similar to the method of adversarial training proposed by Ian Goodfellow Goodfellow et al. (2014). We chose this second method because SGD often did not find a solution with $0$ training error for all the Boolean functions, even with many thousand iterations. By contrast, the adversarial method succeeded in almost all cases, at least for the relatively small neural networks which we focus on here.

We call this method *adversarial SGD*, or *advSGD*, for short. In SGD, the network is trained using the average loss of a random sample of the training set, called a *mini-batch*. In advSGD, after every training step, the classification error for each of the training examples in the mini-batch is computed, and a moving average of each of these classification errors is updated. This moving average gives a score for each training example, measuring how "bad" the network has recently been at predicting this example. Before getting the next mini-batch, the scores of all the examples are passed through a softmax to determine the probability that each example is put in the mini-batch. This way, we force the network to focus on the examples it does worst on. For advSGD, we also used a batch size of 10.

In all experiments we used binary cross entropy as the loss function. We found that Adam could learn Boolean functions with the smaller neural network as well, but only when choosing the mean-squared error loss function.

For the algorithms to approximate the marginal likelihood in Section 5.1, we used a Bernoulli likelihood with a probit link function to approximate the true likelihood (given by the Heaviside function)

## B    TESTING THE APPROXIMATIONS TO THE GAUSSIAN PROCESS MARGINAL LIKELIHOOD

In Figure 5, we show results comparing the empirical frequency of labellings for a sample of $10$ random MNIST images, when these frequencies are obtained by sampling parameters of a neural network (with a Gaussian distribution with parameters $\sigma_w = \sigma_b = 1.0$), versus that calculated using two methods to approximate the marginal likelihood of the Gaussian process corresponding to the neural network architecture we use. We compare the Laplacian and expectation-propagation approximations.(see Rasmussen (2004) for a description of the algorithms) The network has 2 fully connected hidden layers of 784 ReLU neurons each.

## C    THE CHOICE OF VARIANCE HYPERPARAMETERS

One limitation of our approach is that it depends on the choice of the variances of the weights and biases used to define the equivalent Gaussian process. Most of the trends shown in the previous

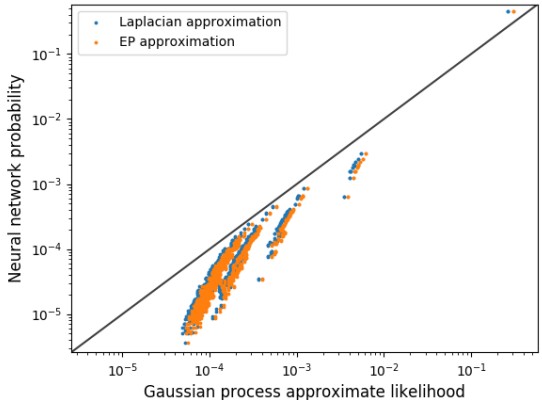

Figure 5: Comparing the empirical frequency of different labellings for a sample of 10 MNIST images obtained from randomly sampling parameters from a neural neural network, versus the approximate marginal likelihood from the corresponding Gaussian process. Orange dots correspond to the expectation-propagation approximation, and blue dots to the Laplace approximation. The network has 2 fully connected hidden layers of 784 ReLU neurons each. The weight and bias variances are 1.0.

section were robust to this choice, but not all. For instance, the bound for MNIST was higher than that for fashion-MNIST for the fully connected network, if the variance was chosen to be 1.0.

In Figures 7 and 6, we show the effect of the variance hyperparameters on the bound. Note that for the fully connected network, the variance of the weights $\sigma_w$ seems to have a much bigger role. This is consistent with what is found in Lee et al. (2017). Furthermore, in Lee et al. (2017) they find, for smaller depths, that the neural network Gaussian process behaves best above $\sigma_w \approx 1.0$, which marks the transition between two phases characterized by the asymptotic behavior of the correlation of activations with depth. This also agrees with the behaviour of the PAC-Bayes bound. For CIFAR10, we find that the bound is best near the phase transition, which is also compatible with results in Lee et al. (2017). For convolutional networks, we found sharper transitions with weight variance, and an larger dependence on bias variance (see Fig. 7). For our experiments, we chose variances values above the phase transition, and which were fixed for each architecture.

The best choice of variance would correspond to the Gaussian distribution best approximates the behaviour of SGD. We measured the variance of the weights after training with SGD and early stopping (stop when 100% accuracy is reached) from a set of initializations, and obtained values an order of magnitude smaller than those used in the experiments above. Using these variances gave significantly worse bounds, above 50% for all levels of corruption.

This measured variance does not necessarily measure the variance of the Gaussian prior that best models SGD, as it also depends on the shape of the zero-error surface (the likelihood function on parameter space). However, it might suggest that SGD is biased towards better solutions in paramater space, giving a stronger/better bias than that predicted only by the parameter-function map with Gaussian sampling of parameters. One way this could happen is if SGD is more likely to find flat (global) "minima"[9] than what is expected from near-uniform sampling of the region of zero-error (probability proportional to volume of minimum). This may be one of the main sources of error in our approach. A better understanding of the relation between SGD and Bayesian parameter sampling is needed to progress on this front (Lee et al. (2017); Matthews et al. (2018)).

---

[9]Note that the notion of minimum is not well defined given that the region of zero error seems to be mostly flat and connected (Sagun et al. (2017); Draxler et al. (2018))

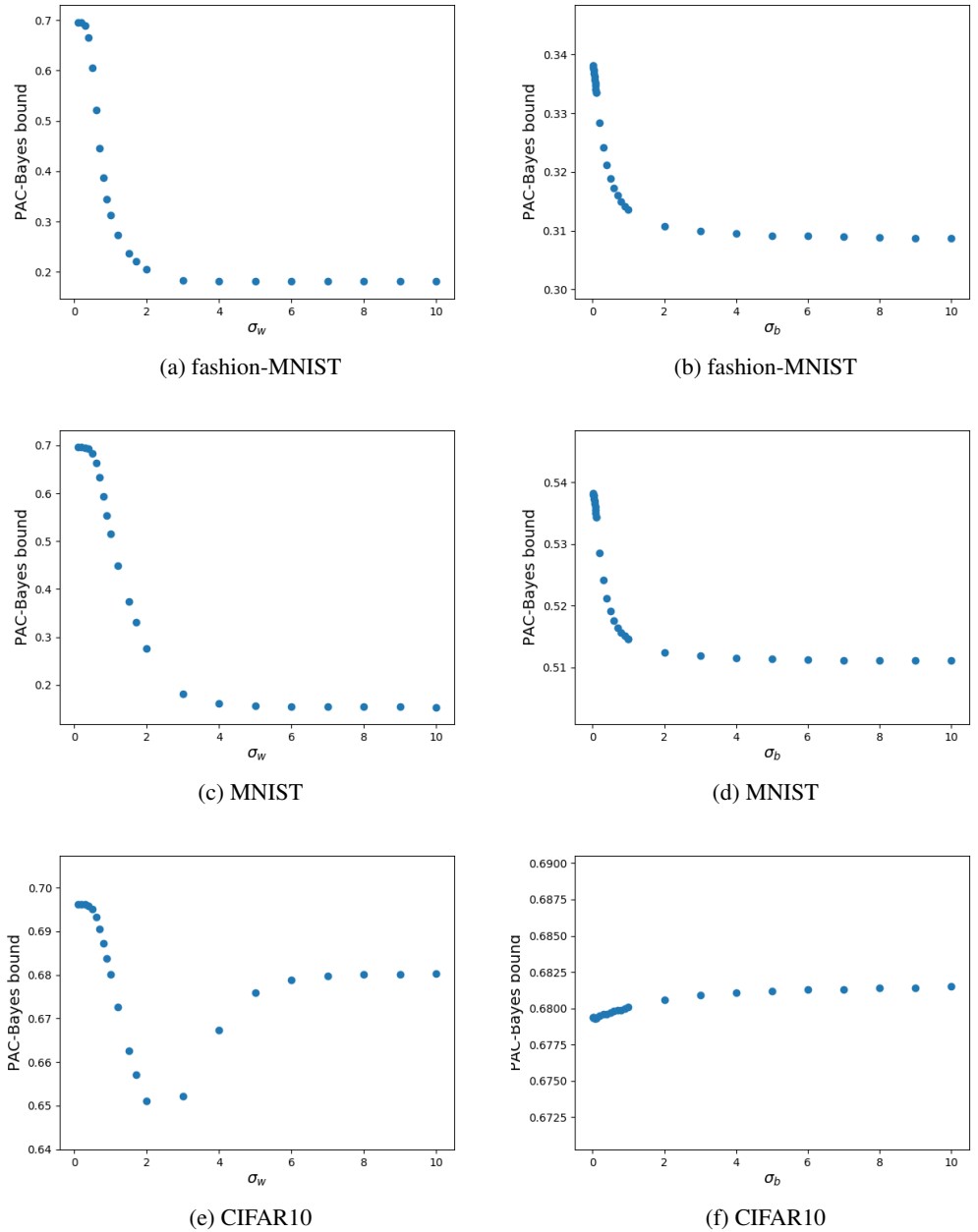

Figure 6: *Dependence of PAC-Bayes bound on variance hyperparameters.* We plot the value of the PAC-Bayes bound versus the standard deviation parameter for the weights and biases, for a sample of 10000 instances from different datasets, and a two-layer fully connected network (with the layers of the same size as input). The fixed parameter is put to $1.0$ in all cases.

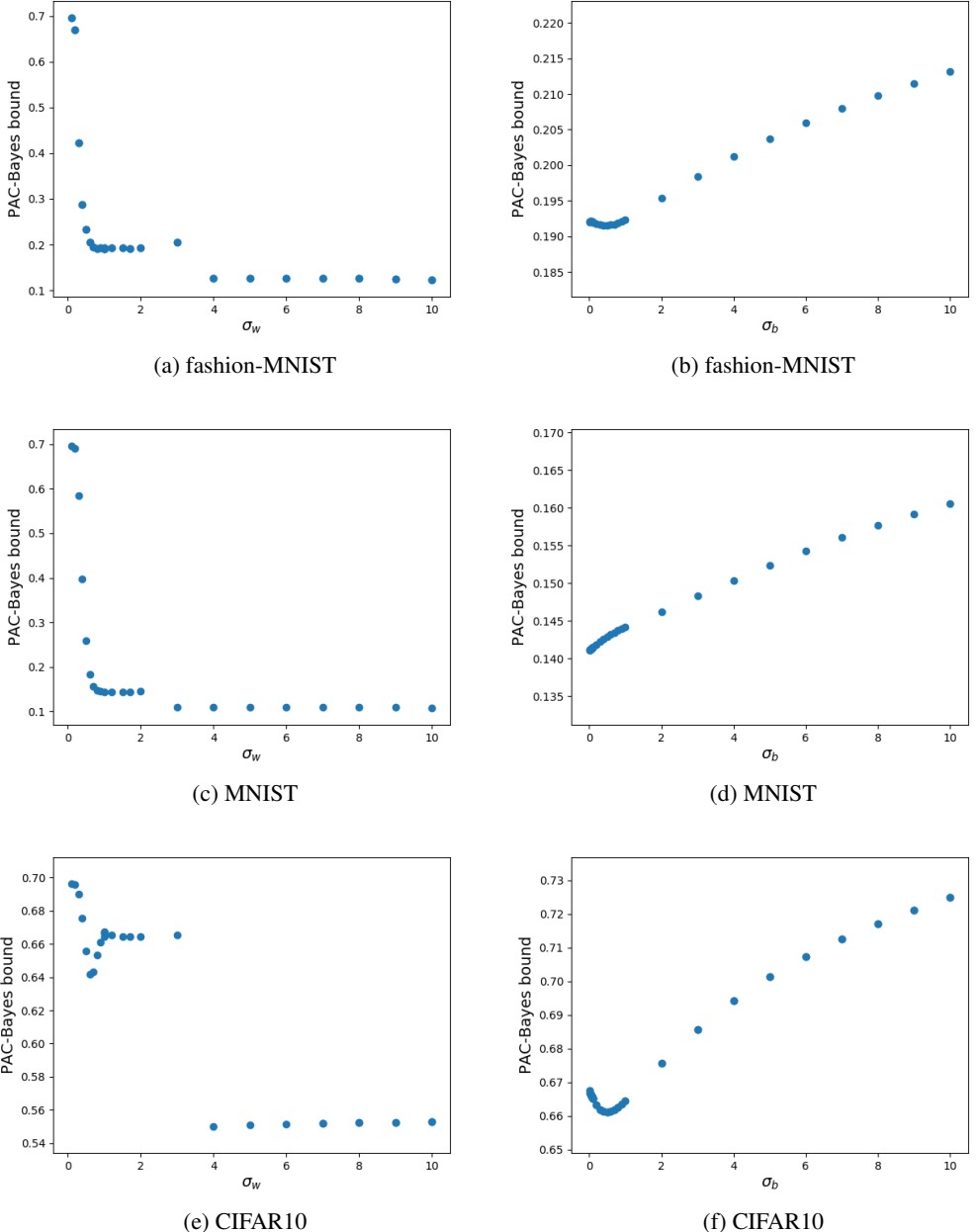

Figure 7: *Dependence of PAC-Bayes bound on variance hyperparameters.* PAC-Bayes bound versus the standard deviation parameter for the weights and biases, for a sample of 10000 instances from different datasets, and a four-layer convolutional network. The fixed parameter is put to 1.0 in all cases.

## D  DETAILS ON THE EXPERIMENTS COMPARING THE PROBABILITY OF FINDING A FUNCTION BY SGD AND NEURAL NETWORK GAUSSIAN PROCESSES

Here we describe in more detail the experiments carried out in Section 7 in the main text. We aim to compare the probability $P(f|S)$ of finding a particular function $f$ when training with SGD, and when training with approximate Bayesian inference (ABI), given a training set $S$. We consider this probability, averaged over of training sets, to look at the properties of the algorithm rather than some particular training set. In particular, consider a sample of $N$ training sets $\{S_1, ..., S_N\}$ of size 118. Then we are interested in the quantity (which we refer to as *average probability of finding function $f$*):

$$\langle P(f) \rangle := \frac{1}{N} \sum_{i=1}^{n} P(f|S_i)$$

For the SGD-like algorithms, we approximate $P(f|S_i)$ by the fraction of times we obtain $f$ from a set of $M$ random runs of the algorithm (with random initializations too). For ABI, we calculate it as:

$$P(f|S_i) = \begin{cases} \frac{P(f)}{P(S_i)} & \text{if } f \text{ consistent with } S_i \\ 0 & \text{otherwise} \end{cases},$$

where $P(f)$ is the prior probability of $f$ computed using the EP approximation of the Gaussian process corresponding to the architecture we use, and $P(S_i)$ is the marginal likelihood of $S_i$ under this prior probability.

In Fig. 8, we show results for the same experiment as in Figure 4 in main text, but with lower variance hyperparameters for the Gaussian process, which results in significantly worse correlation.

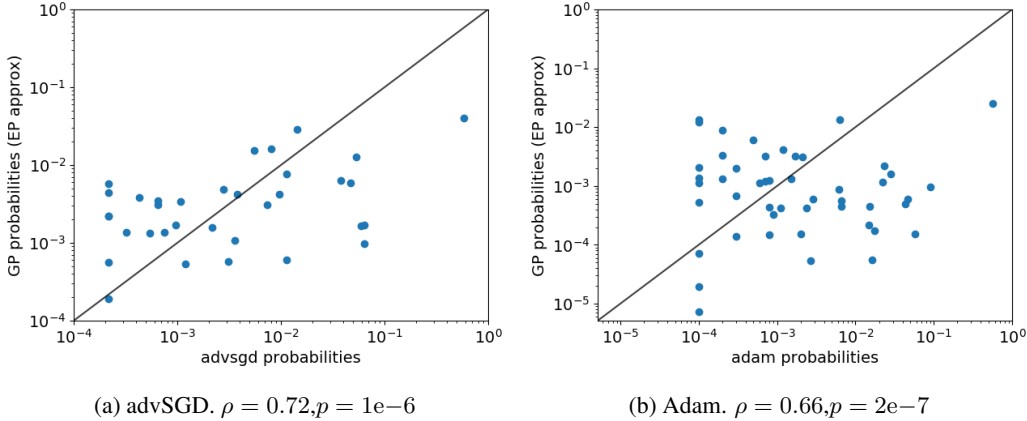

(a) advSGD. $\rho = 0.72, p = 1\mathrm{e}{-6}$          (b) Adam. $\rho = 0.66, p = 2\mathrm{e}{-7}$

Figure 8: We plot $\langle P(f) \rangle$ for a variant of SGD, versus $\langle P(f) \rangle$ computed using the ABI method described above (which approximates uniform sampling on parameter space on the zero-error region). The Gaussian process parameters (see Section 5.1 in main text) are $\sigma_w = 1.0$, and $\sigma_b = 1.0$. For advSGD, we have removed functions which only appared once in the whole sample, to avoid finite-size effects. In the captions $\rho$ refers to the 2-tailed Pearson correlation coefficient, and $p$ to its corresponding p value.

## E  SIMPLICITY BIAS AND THE PARAMETER-FUNCTION MAP

An important argument in Section 3 in the main text is that the parameter-function map of neural networks should exhibit the basic simplicity bias phenomenlgy recently described in Dingle et al.

in Dingle et al. (2018). In this section we briely describe some key results of reference Dingle et al. (2018) relevant to this argument.

A computable[10] input-output map $f : I \rightarrow O$, mapping $N_I$ inputs from the set $I$ to $N_O$ outputs $x$ from the set $O$[11] may exhibit simplicity bias if the following restrictions are satisfied (Dingle et al. (2018)):

1) *Map simplicity* The map should have limited complexity, that is its Kolmogorov complexity $K(f)$ should asymptotically satisfy $K(f) + K(n) \ll K(x) + O(1)$, for typical $x \in O$ where $n$ is a measure of the size of the input set (e.g. for binary input sequences, $N_I = 2^n$.).

2) *Redundancy*: There should be many more inputs than outputs ($N_I \gg N_O$) so that the probability $P(x)$ that the map generates output $x$ upon random selection of inputs $\in I$ can in principle vary significantly.

3) *Finite size* $N_O \gg 1$ to avoid potential finite size effects.

4) *Nonlinearity*: The map $f$ must be a nonlinear function since linear functions do not exhibit bias.

5) *Well behaved*: The map should not primarily produce pseudorandom outputs (such as the digits of $\pi$), because complexity approximators needed for practical applications will mistakenly label these as highly complex.

For the deep learning learning systems studied in this paper, the inputs of the map $f$ are the parameters that fix the weights for the particular neural network architecture chosen, and the outputs are the functions that the system produces. Consider, for example, the configuration for Boolean functions studied in the main text. While the output functions rapidly grow in complexity with increasing size of the input layer, the map itself can be described with a low-complexity procedure, since it consists of reading the list of parameters, populating a given neural network architecture and evaluating it for all inputs. For reasonable architectures, the information needed to describe the map grows logarithmically with the input dimension $n$, so for large enough $n$, the amount of information required to describe the map will be much less than the information needed to describe a typical function, which requires $2^{2^n}$ bits. Thus the Kolmogorov complexity $K(f)$ of this map is asymptotically smaller than the the typical complexity of the output, as required by the map simplicity condition 1) above.

The redundancy condition 2) depends on the network architecture and discretization. For overparameterised networks, this condition is typically satisfied. In our specific case, where we use floating point numbers for the parameters (input set $I$), and Boolean functions (output set $O$), this condition is clearly satisfied. Neural nets can represent very large numbers of potential functions (see for example estimates of VC dimension Harvey et al. (2017); Baum & Haussler (1989)), so that condition 3) is also generally satisfied. Neural network parameter-function maps are evidently non-linear, satisfying condition 4). Condition 5) is perhaps the least understood condition within simplicity bias. However, the lack of any function with high probability and high complexity (at least when using LZ complexity), provides some empirical validation. This condition also agrees with the expectation that neural networks will not predict the outputs of a good pseudorandom number generator. One of the implicit assumptions in the simplicity bias framework is that, although true Kolmogorov complexity is always uncomputable, approximations based on well chosen complexity measures perform well for most relevant outputs $x$. Nevertheless, where and when this assumptions holds is a deep problem for which further research is needed.

## F    OTHER COMPLEXITY MEASURES

One of the key steps to practical application of the simplicity bias framework of Dingle et al. in Dingle et al. (2018) is the identification of a suitable complexity measure $\tilde{K}(x)$ which mimics aspects of the (uncomputable) Kolmogorov complexity $K(x)$ for the problem being studied. It was shown for

---

[10]Here computable simply means that all inputs lead to outputs, in other words there is no halting problem.

[11]This language of finite input and outputs sets assumes discrete inputs and outputs, either because they are intrinsically discrete, or because they can be made discrete by a coarse-graining procedure. For the parameter-function maps studied in this paper the set of outputs (the full hypothesis class) is typically naturally discrete, but the inputs are continuous. However, the input parameters can always be discretised without any loss of generality.

the maps in Dingle et al. (2018) that several different complexity measures all generated the same qualitative simplicity bias behaviour:

$$P(x) \leq 2^{-(a\tilde{K}(x)+b)} \tag{4}$$

but with different values of $a$ and $b$ depending on the complexity measure and of course depending on the map, but independent of output $x$. Showing that the same qualitative results obtain for different complexity measures is sign of robustness for simplicity bias.

Below we list a number of different descriptional complexity measures which we used, to extend the experiments in Section 3 in the main text.

### F.1 COMPLEXTY MEASURES

*Lempel-Ziv complexity* (*LZ complexity* for short). The Boolean functions studied in the main text can be written as binary strings, which makes it possible to use measures of complexity based on finding regularities in binary strings. One of the best is Lempel-Ziv complexity, based on the Lempel-Ziv compression algorithm. It has many nice properties, like asymptotic optimality, and being asymptotically equal to the Kolmogorov complexity for an ergodic source. We use the variation of Lempel-Ziv complexity from Dingle et al. (2018) which is based on the 1976 Lempel Ziv algorithm (Lempel & Ziv (1976)):

$$K_{LZ}(x) = \begin{cases} \log_2(n), & x = 0^n \text{ or } 1^n \\ \log_2(n)[N_w(x_1...x_n) + N_w(x_n...x_1)]/2, & \text{otherwise} \end{cases} \tag{5}$$

where $n$ is the length of the binary string, and $N_w(x_1...x_n)$ is the number of words in the Lempel-Ziv "dictionary" when it compresses output $x$. The symmetrization makes the measure more fine-grained, and the $\log_2(n)$ factor as well as the value for the simplest strings ensures that they scale as expected for Kolmogorov complexity. This complexity measure is the primary one used in the main text.

We note that the binary string representation depends on the order in which inputs are listed to construct it, which is not a feature of the function itself. This may affect the LZ complexity, although for low-complexity input orderings (we use numerical ordering of the binary inputs), it has a negligible effect, so that $K(x)$ will be very close to the Kolmogorov complexity of the function.

*Entropy*. A fundamental, though weak, measure of complexity is the entropy. For a given binary string this is defined as $S = -\frac{n_0}{N}\log_2\frac{n_0}{N} - \frac{n_1}{N}\log_2\frac{n_1}{N}$, where $n_0$ is the number of zeros in the string, and $n_1$ is the number of ones, and $N = n_0 + n_1$. This measure is close to 1 when the number of ones and zeros is similar, and is close to 0 when the string is mostly ones, or mostly zeros. Entropy and $K_{LZ}(x)$ are compared in Fig. 9, and in more detail in supplementary note 7 (and supplementary information figure 1) of reference Dingle et al. (2018). They correlate, in the sense that low entropy $S(x)$ means low $K_{LZ}(x)$, but it is also possible to have Large entropy but low $K_{LZ}(x)$, for example for a string such as 10101010....

*Boolean expression complexity*. Boolean functions can be compressed by finding simpler ways to represent them. We used the standard SciPy implementation of the Quine-McCluskey algorithm to minimize the Boolean function into a small sum of products form, and then defined the number of operations in the resulting Boolean expression as a *Boolean complexity* measure.

*Generalization complexity*. L. Franco et al. have introduced a complexity measure for Boolean functions, designed to capture how difficult the function is to learn and generalize (Franco & Anthony (2004)), which was used to empirically find that simple functions generalize better in a neural network (Franco (2006)). The measure consists of a sum of terms, each measuring the average over all inputs fraction of neighbours which change the output. The first term considers neighbours at Hamming distance of 1, the second at Hamming distance of 2 and so on. The first term is also known (up to a normalization constant) as average sensitivity (Friedgut (1998)). The terms in the series have also been called "generalized robustness" in the evolutionary theory literature (Greenbury et al. (2016)). Here we use the first two terms, so the measure is:

$$C(f) = C_1(f) + C_2(f),$$

$$C_1(f) = \frac{1}{2^n n} \sum_{x \in X} \sum_{y \in \text{Nei}_1(x)} |f(x) - f(y)|,$$

$$C_1(f) = \frac{2}{2^n n(n-1)} \sum_{x \in X} \sum_{y \in \text{Nei}_2(x)} |f(x) - f(y)|,$$

where $\text{Nei}_i(x)$ is all neighbours of $x$ at Hamming distance $i$.

*Critical sample ratio*. A measure of the complexity of a function was introduced in Krueger et al. (2017) to explore the dependence of generalization with complexity. In general, it is defined with respect to a sample of inputs as the fraction of those samples which are *critical samples*, defined to be an input such that there is another input within a ball of radius $r$, producing a different output (for discrete outputs). Here, we define it as the fraction of all inputs, that have another input at Hamming distance 1, producing a different output.

## F.2 CORRELATION BETWEEN COMPLEXITIES

In Fig. 9, we compare the different complexity measures against one another. We also plot the frequency of each complexity; generally more functions are found with higher complexity.

## F.3 PROBABILITY-COMPLEXITY PLOTS

In Fig. 10 we show how the probability versus complexity plots look for other complexity measures. The behaviour is similar to that seen for the LZ complexity measure in Fig 1(b) of the main text. In Fig. 12 we show probability versus LZ complexity plots for other choices of parameter distributions.

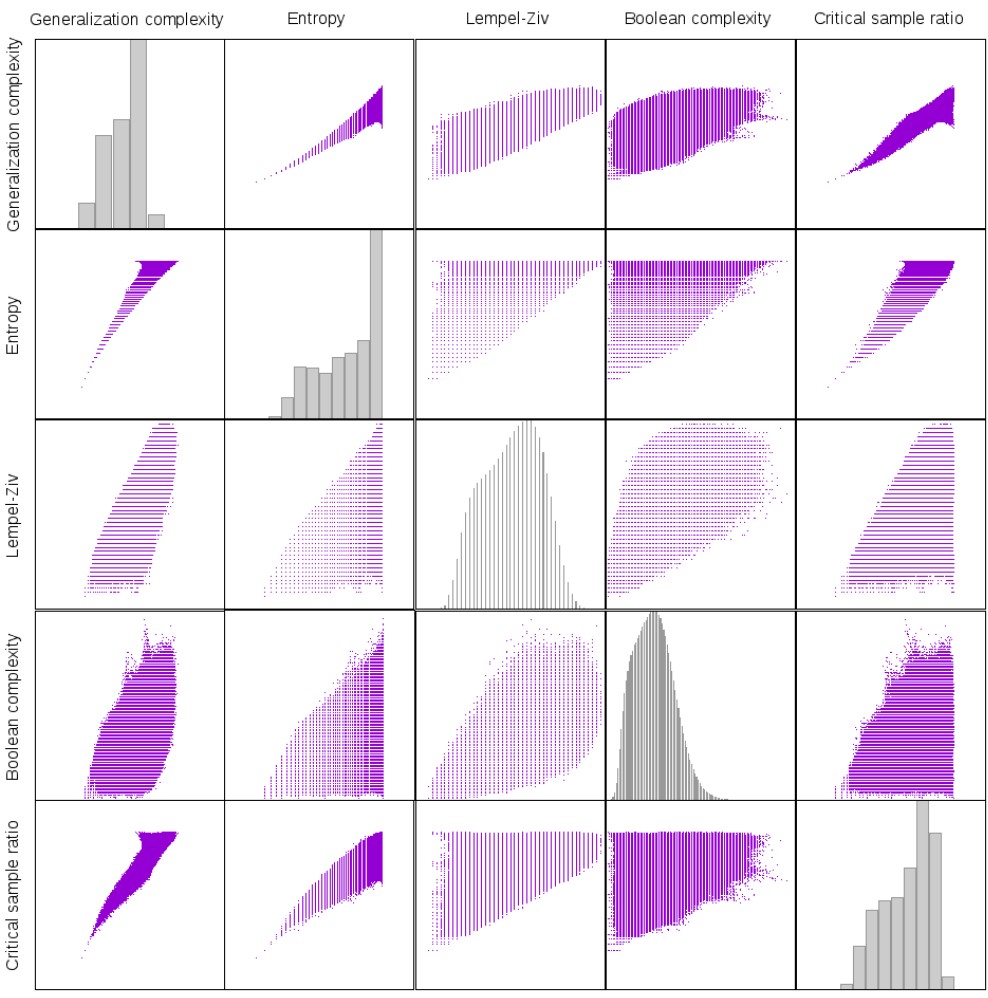

Figure 9: Scatter matrix showing the correlation between the different complexity measures used in this paper On the diagonal, a histogram (in grey) of frequency versus complexity is depicted. The functions are from the sample of $10^8$ parameters for the $(7, 40, 40, 1)$ network.

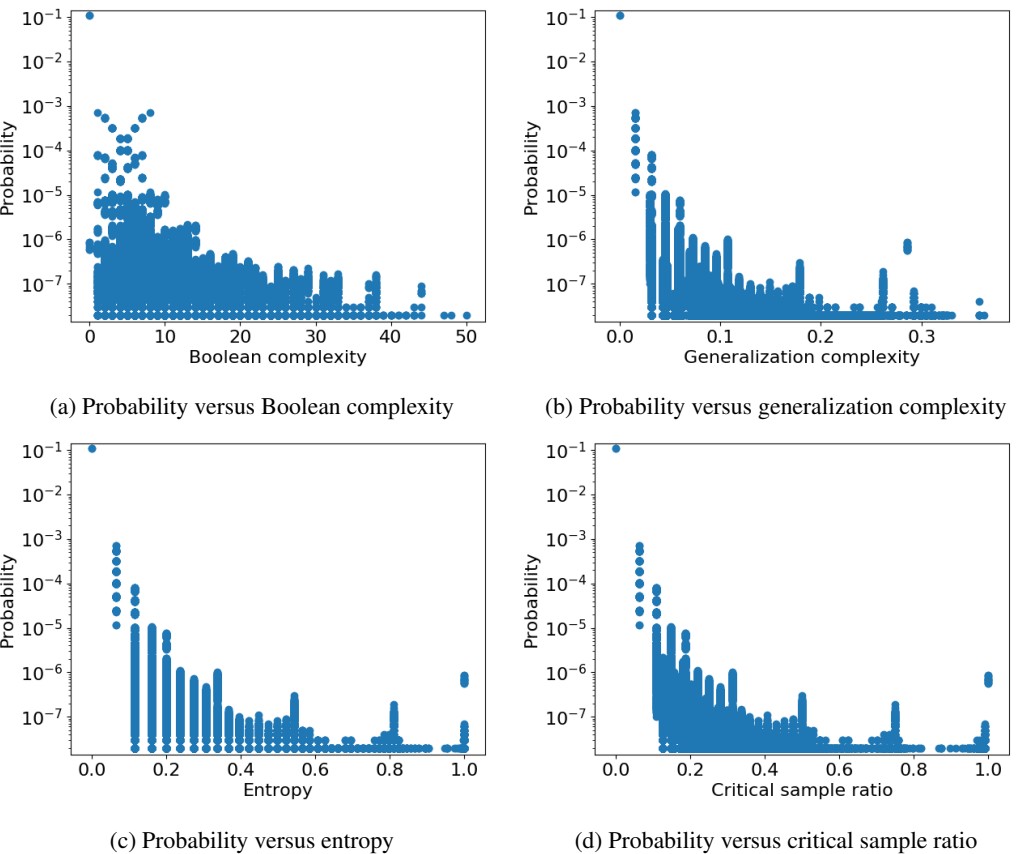

(a) Probability versus Boolean complexity

(b) Probability versus generalization complexity

(c) Probability versus entropy

(d) Probability versus critical sample ratio

Figure 10: Probability versus different measures of complexity (see main text for Lempel-Ziv), estimated from a sample of $10^8$ parameters, for a network of shape $(7, 40, 40, 1)$. Points with a frequency of $10^{-8}$ are removed for clarity because these suffer from finite-size effects (see Appendix G). The measures of complexity are described in Appendix F.

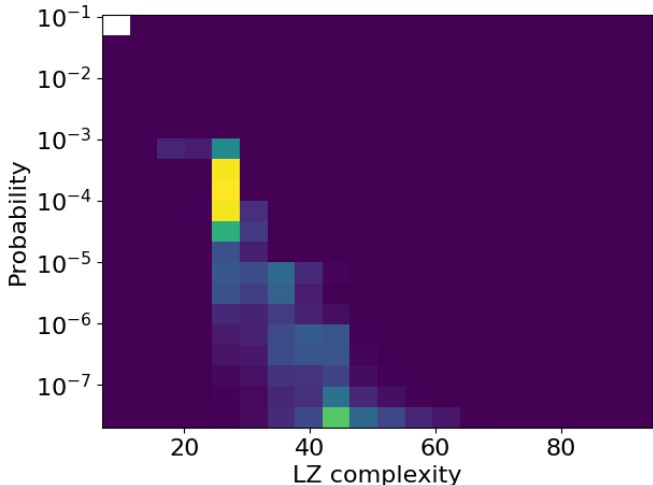

Figure 11: Histogram of functions in the probability versus Lempel-Ziv complexity plane, weighted according to their probability. Probabilities are estimated from a sample of $10^8$ parameters, for a network of shape $(7, 40, 40, 1)$

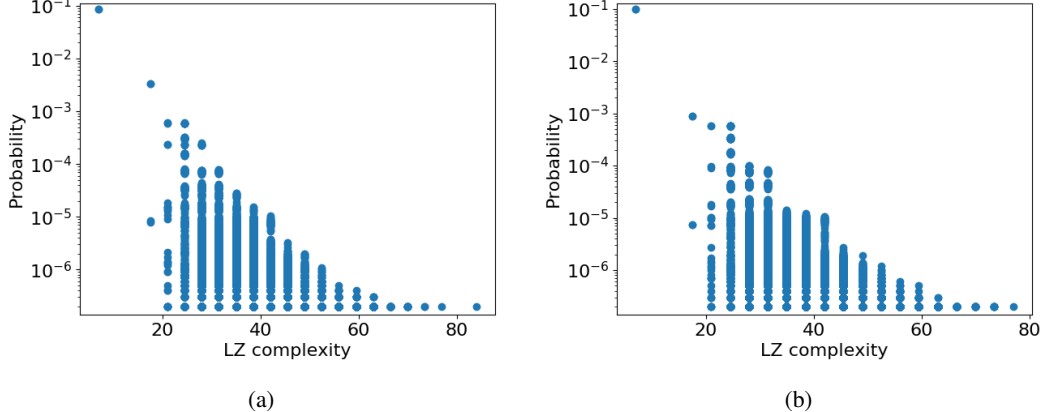

(a)                                                    (b)

Figure 12: Probability versus LZ complexity for network of shape $(7, 40, 40, 1)$ and varying sampling distributions. Samples are of size $10^7$. (a) Weights are sampled from a Gaussian with variance $1/\sqrt{n}$ where $n$ is the input dimension of each layer. (b) Weights are sampled from a Gaussian with variance $2.5$

### F.4 Effects of target function complexity on learning for different complexity measures

Here we show the effect of the complexity of the target function on learning, as well as other complementary results. Here we compare neural network learning to random guessing, which we call "unbiased learner". Note that both probably have the same hypothesis class as we tested that the neural network used here can fit random functions.

The functions in these experiments were chosen by randomly sampling parameters of the neural network used, and so even the highest complexity ones are probably not fully random[12]. In fact, when training the network on truly random functions, we obtain generalization errors equal or above those of the unbiased learner. This is expected from the No Free Lunch theorem, which says that no algorithm can generalize better (for off-training error) uniformly over all functions than any other algorithm (Wolpert & Waters (1994)).

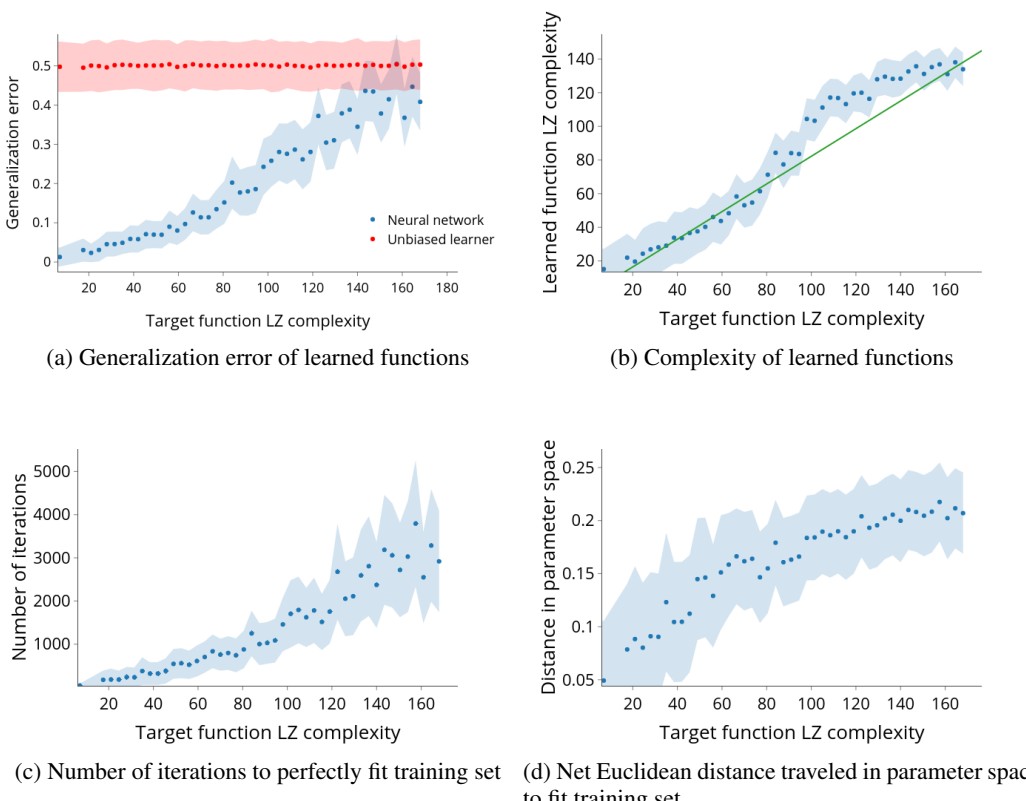

(a) Generalization error of learned functions

(b) Complexity of learned functions

(c) Number of iterations to perfectly fit training set

(d) Net Euclidean distance traveled in parameter space to fit training set

Figure 13: Different learning metrics versus the LZ complexity of the target function, when learning with a network of shape $(7, 40, 40, 1)$. Dots represent the means, while the shaded envelope corresponds to piecewise linear interpolation of the standard deviation, over 500 random initializations and training sets.

### F.5 Lempel-Ziv versus Entropy

To check that the correlation between LZ complexity and generalization is not only because of a correlation with function entropy (which is just a measure of the fraction of inputs mapping to

---

[12]The fact that non-random strings can have maximum LZ complexity is a consequence of LZ complexity being a less powerful complexity measure than Kolmogorov complexity, see e.g. Estevez-Rams et al. (2013). The fact that neural networks do well for non-random functions, even if they have maximum LZ, suggests that their simplicity bias captures a notion of complexity stronger than LZ.

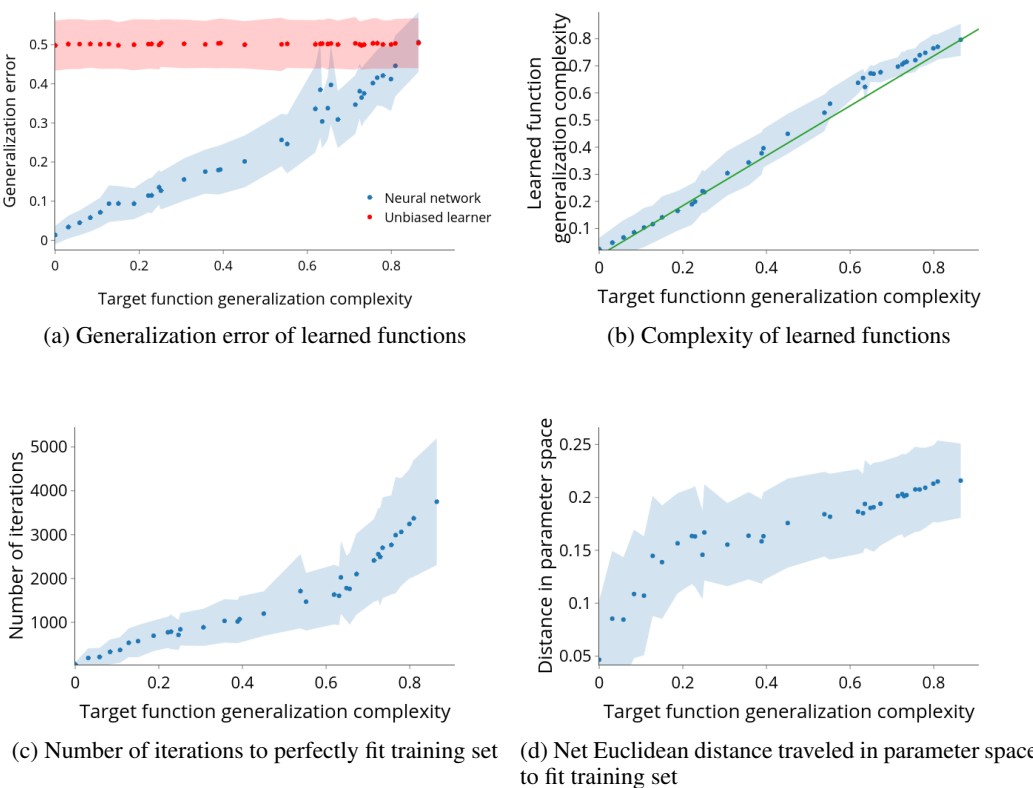

(a) Generalization error of learned functions

(b) Complexity of learned functions

(c) Number of iterations to perfectly fit training set

(d) Net Euclidean distance traveled in parameter space to fit training set

Figure 14: Different learning metrics versus the generalization complexity of the target function, when learning with a network of shape $(7, 40, 40, 1)$. Dots represent the means, while the shaded envelope corresponds to piecewise linear interpolation of the standard deviation, over $500$ random initializations and training sets.

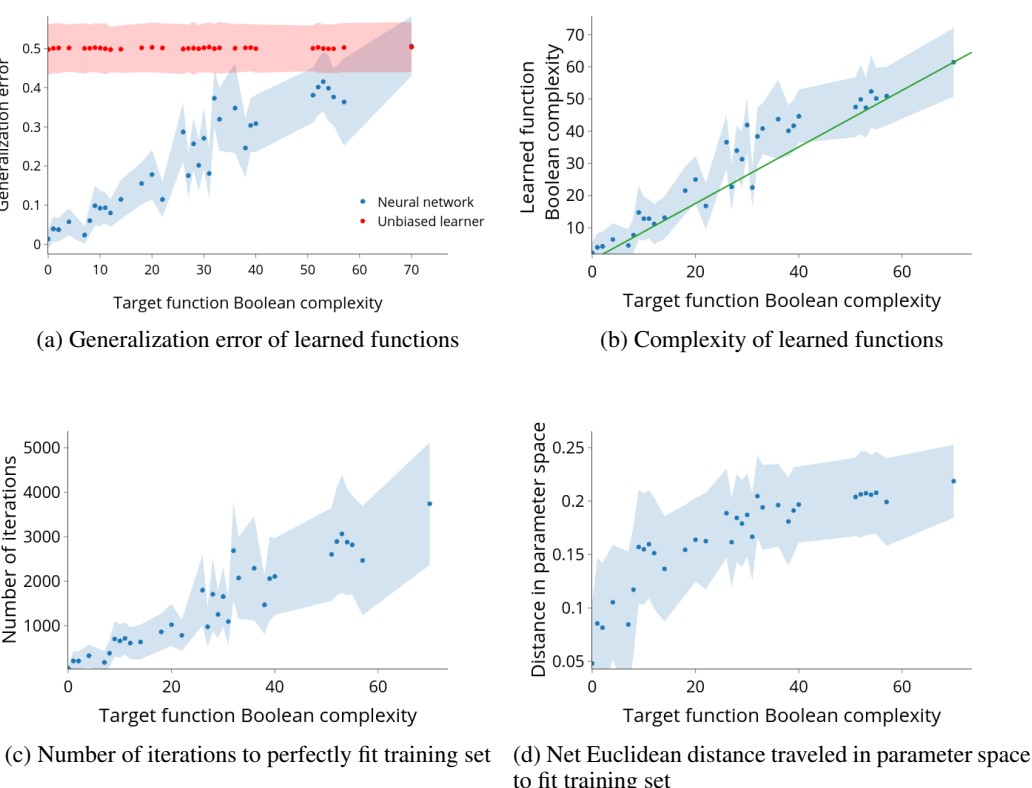

(a) Generalization error of learned functions

(b) Complexity of learned functions

(c) Number of iterations to perfectly fit training set

(d) Net Euclidean distance traveled in parameter space to fit training set

Figure 15: Different learning metrics versus the Boolean complexity of the target function, when learning with a network of shape $(7, 40, 40, 1)$. Dots represent the means, while the shaded envelope corresponds to piecewise linear interpolation of the standard deviation, over $500$ random initializations and training sets.

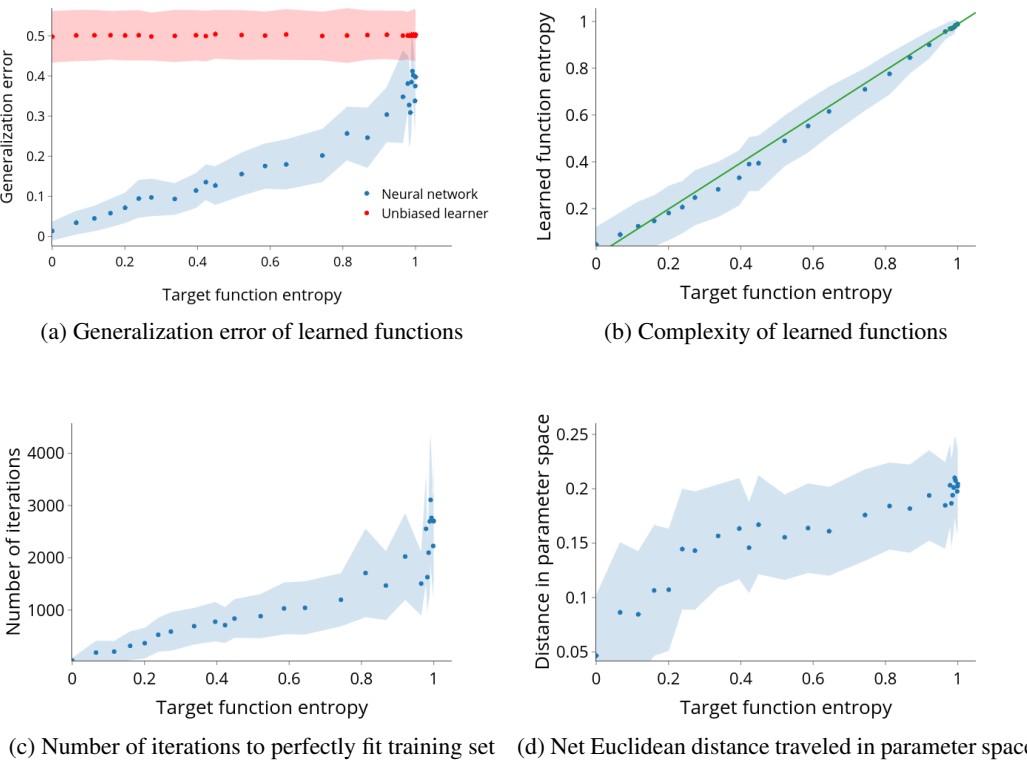

(a) Generalization error of learned functions

(b) Complexity of learned functions

(c) Number of iterations to perfectly fit training set

(d) Net Euclidean distance traveled in parameter space to fit training set

Figure 16: Different learning metrics versus the entropy of the target function, when learning with a network of shape $(7, 40, 40, 1)$. Dots represent the means, while the shaded envelope corresponds to piecewise linear interpolation of the standard deviation, over $500$ random initializations and training sets.

1 or 0, see Section F), we observed that for some target functions with maximum entropy (but which are simple when measured using LZ complexity), the network still generalizes better than the unbiased learner, showing that the bias towards simpler functions is better captured by more powerful complexity measures than entropy[13]. This is confirmed by the results in Fig. 17 where we fix the target function entropy to 1.0, and observe that the generalization error still exhibits considerable variation, as well as a positive correlation with complexity

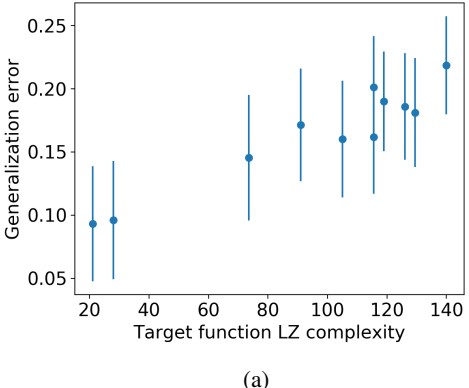 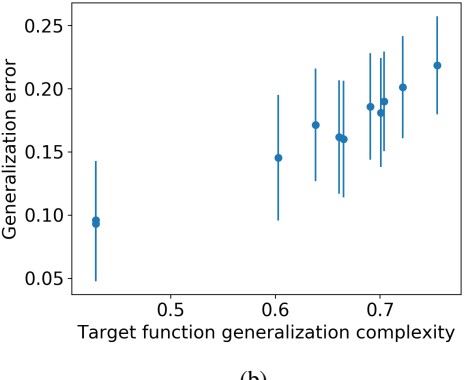

(a)                                        (b)

Figure 17: Generalization error of learned function versus the complexity of the target function for target functions with fixed entropy 1.0, for a network of shape $(7, 20, 20, 1)$. Complexity measures are (a) LZ and (b) generalisation complexity. Here the training set size was of size 64, but sampled with replacement, and the generalization error is over the whole input space. Note that despite the fixed entropy there is still variation in generalization error, which correlates with the complexity of the function. These figures demonstrate that entropy is a less accurate complexity measure than LZ or generalisation complexity, for predicting generalization performance.

## G    FINITE-SIZE EFFECTS FOR SAMPLING PROBABILITY

Since for a sample of size $N$ the minimum estimated probability is $1/N$, many of the low-probability samples that arise just once may in fact have a much lower probability than suggested. See Figure 18), for an illustration of how this finite-size sampling effect manifests with changing sample size $N$. For this reason, these points are typically removed from plots.

## H    EFFECT OF NUMBER OF LAYERS ON SIMPLICITY BIAS

In Figure 19 we show the effect of the number of layers on the bias (for feedforward neural networks with 40 neurons per layer). The left figures show the probability of individual functions versus the complexity. The right figure shows the histogram of complexities, weighted by the probability by which the function appeared in the sample of parameters. The histograms therefore show the distribution over complexities when randomly sampling parameters[14] We can see that between the 0 layer perceptron and the 2 layer network there is an increased number of higher complexity functions. This is most likely because of the increasing expressivity of the network. For 2 layers and above, the expressivity does not significantly change, and instead, we observe a shift of the distribution towards lower complexity.

---

[13]LZ is a better approximation to Kolmogorov complexity than entropy (Cover & Thomas (2012)), but of course LZ can still fail, for example when measuring the complexity of the digits of $\pi$.

[14]using a Gaussian with $1/sqrtn$ variance in this case, $n$ being number of inputs to neuron

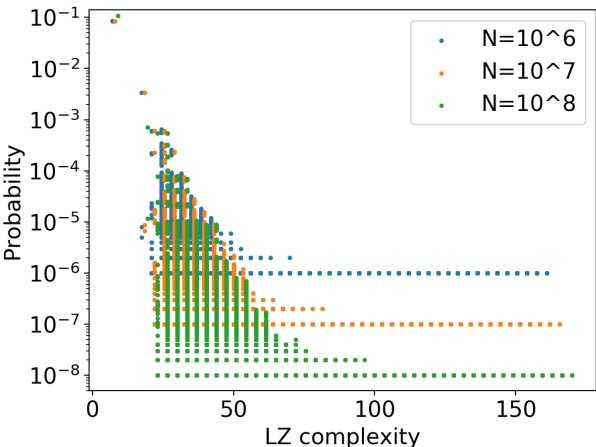

Figure 18: Probability (calculated from frequency) versus Lempel-Ziv complexity for a neural network of shape $(7, 40, 40, 1)$, and sample sizes $N = 10^6, 10^7, 10^8$. The lowest frequency functions for a given sample size can be seen to suffer from finite-size effects, causing them to have a higher frequency than their true probability.

## I  BIAS AND THE CURSE OF DIMENSIONALITY

We have argued that the main reason deep neural networks are able to generalize is because their implicit prior over functions is heavily biased. We base this claim on PAC-Bayes theory, which tells us that enough bias implies generalization. The contrapositive of this claim is that bad generalization implies small bias. In other words, models which perform poorly on certain tasks relative to deep learning, should show little bias. Here we describe some examples of this, connecting them to the curse of dimensionality. In future work, we plan to explore the converse statement, that small bias implies bad generalization — one way of approaching this would be via lower bounds matching the PAC-Bayes upper bound.

Complex machine learning tasks require models which are expressive enough to be able to learn the target function. For this reason, before deep learning, the main approach to complex tasks was to use non-parametric models which are infinitely expressible. These include Gaussian processes and other kernel methods. However, unlike deep learning, these models were not successful when applied to tasks where the dimensionality of the input space was very large. We therefore expect that these models show little bias, as they generalize poorly.

Many of these models use kernels which encode some notion of local continuity. For example, the Gaussian kernel ensures that points within a ball of radius $\lambda$ are highly correlated. On the other hand, points separated by a distance greater than $\lambda$ can be very different. Intuitively, we can divide the space into regions of length scale $\lambda$. If the input domain we re considering has $O(1)$ volume, and has dimensionality $d$ (is a subset of $\mathbb{R}^d$), then the volume of each of these regions is of order $\lambda^d$, and the number of these regions is of order $1/\lambda^d$. In the case of binary classification, we can estimate the effective number of functions which the kernel "prefers" by constraining the function to take label 0 or 1 within each region, but with no further constraint. The number of such functions is $2^{a^d}$, where we let $a := 1/\lambda$. Each of these functions is equally likely, and together they take the bulk of the total probability, so that they have probability close to $2^{-a^d}$, which decreases very quickly with dimension.

Kernels like the Gaussian kernel are biased towards functions which are locally continuous. However, for high dimension $d$, they are not biased *enough*. In particular, as the the probability of the most likely functions grows doubly exponentially with $d$, we expect PAC-Bayes-like bounds to grow exponentially with $d$, quickly becoming vacuous. This argument is essentially a way of understanding the curse of dimensionality from the persective of priors over functions.

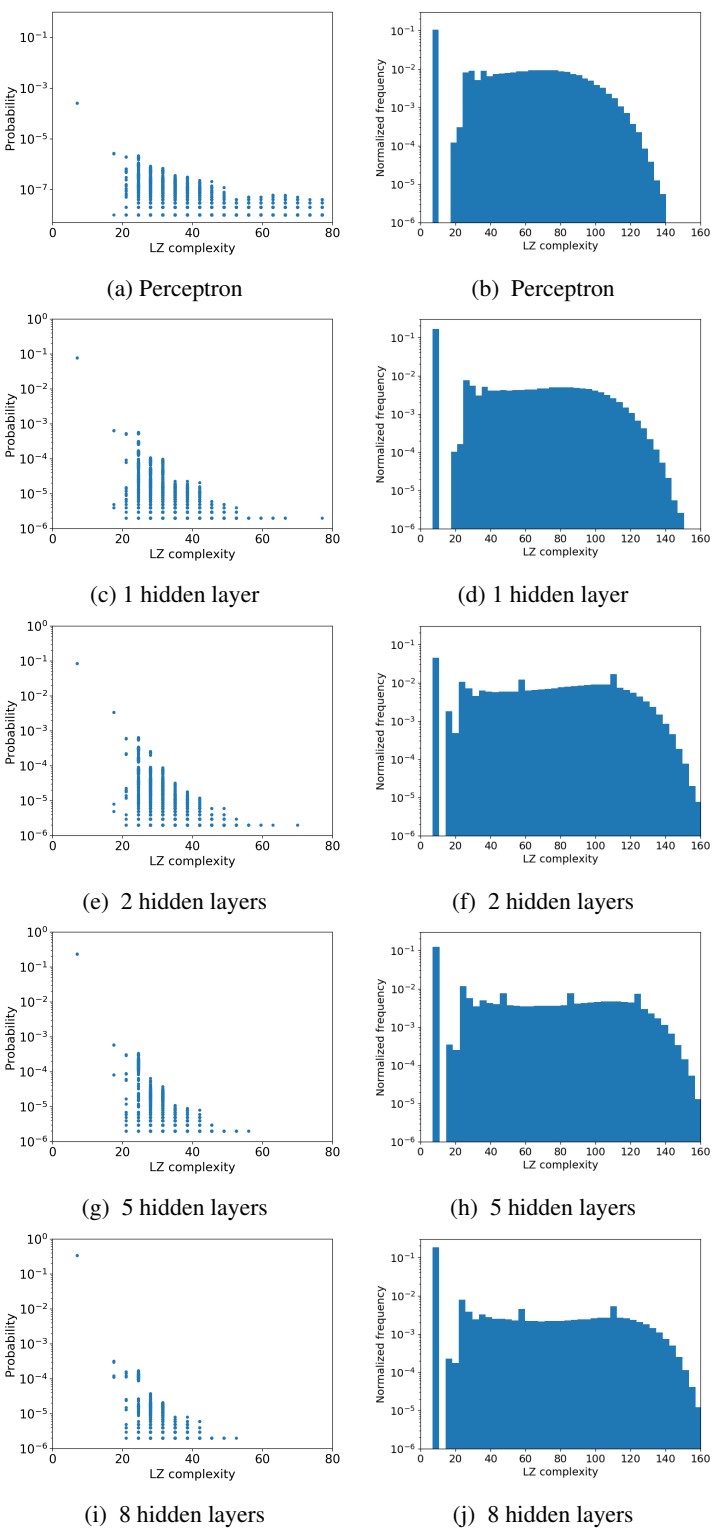

Figure 19: Probability versus LZ complexity for networks with different number of layers. Samples are of size $10^6$ (a) & (b) A perceptron with 7 input neurons (complexity is capped at 80 in (a) to aid comparison with the other figures). (c) & (d) A network with 1 hidden layer of 40 neurons (e) & (f) A network with 2 hidden layer of 40 neurons (g) & (h) A network with 5 hidden layers of 40 neurons each. (i) & (j) A network with 8 hidden layers of 40 neurons each

## J    OTHER RELATED WORK

The topic of generalization in neural networks has been extensively studied both in theory and experiment, and the literature is vast. Theoretical approaches to generalization include classical notions like VC dimension Baum & Haussler (1989); Harvey et al. (2017) and Rademacher complexity Sun et al. (2016), but also more modern concepts such as stability Hardt et al. (2016), robustness Xu & Mannor (2012), compression Arora et al. (2018) as well as studies on the relation between generalization and properties of stochastic gradient descent (SGD) algorithms Zhang et al. (2017b); Soudry et al. (2017); Advani & Saxe (2017).

Empirical studies have also pushed the boundaries proposed by theory, In particular, in recent work by Zhang et al. Zhang et al. (2017a), it is shown that while deep neural networks are expressive enough to fit randomly labeled data, they can still generalize for data with structure. The generalization error correlates with the amount of randomization in the labels. A similar result was found much earlier in experiments with smaller neural networks Franco (2006), where the authors defined a complexity measure for Boolean functions, called generalization complexity (see Appendix F), which appears to correlate well with the generalization error.

Inspired by the results of Zhang et al. Zhang et al. (2017a), Arpit et al. Krueger et al. (2017) propose that the data dependence of generalization for neural networks can be explained because they tend to prioritize learning simple patterns first. The authors show some experimental evidence supporting this hypothesis, and suggest that SGD might be the origin of this implicit regularization. This argument is inspired by the fact that SGD converges to minimum norm solutions for linear models Yao et al. (2007), but only suggestive empirical results are available for the case of nonlinear models, so that the question remains open Soudry et al. (2017). Wu et al. Wu et al. (2017) argue that full-batch gradient descent also generalizes well, suggesting that SGD is not the main cause behind generalization. It may be that SGD provides some form of implicit regularisation, but here we argue that the exponential bias towards simplicity is so strong that it is likely the main origin of the implicit regularization in the parameter-function map.

The idea of having a bias towards simple patterns has a long history, going back to the philosophical principle of Occam's razor, but having been formalized much more recently in several ways in learning theory. For instance, the concepts of minimum description length (MDL) Rissanen (1978), Blumer algorithms Blumer et al. (1987); Wolpert & Waters (1994), and universal induction Ming & Vitányi (2014) all rely on a bias towards simple hypotheses. Interestingly, these approaches go hand in hand with non-uniform learnability, which is an area of learning theory which tries to predict data-dependent generalization. For example, MDL tends to be analyzed using structural risk minimization or the related PAC-Bayes approach Vapnik (2013); Shalev-Shwartz & Ben-David (2014).

Hutter et al. Lattimore & Hutter (2013) have shown that the generalization error grows with the target function complexity for a perfect Occam algorithm[15] which uses Kolmogorov complexity to choose between hypotheses. Schmidhuber applied variants of universal induction to learn neural networks Schmidhuber (1997). The simplicity bias from Dingle et al. Dingle et al. (2018) arises from a simpler version of the coding theorem of Solomonoff and Levin Ming & Vitányi (2014). More theoretical work is needed to make these connections rigorous, but it may be that neural networks intrinsically approximate universal induction because the parameter-function map results in a prior which approximates the universal distribution.

Other approaches that have been explored for neural networks try to bound generalization by bounding capacity measures like different types of norms of the weights (Neyshabur et al. (2015); Keskar et al. (2016); Neyshabur et al. (2017b;a); Bartlett et al. (2017); Golowich et al. (2018); Arora et al. (2018)), or unit capacity Neyshabur et al. (2018). These capture the behaviour of the real test error (like its improvement with overparametrization (Neyshabur et al. (2018)), or with training epoch (Arora et al. (2018))). However, these approaches have not been able to obtain nonvacuous bounds yet.

---

[15]Here what we call a 'perfect Occam algorithm' is an algorithm which returns the simplest hypothesis which is consistent with the training data, as measured using some complexity measure, such as Kolmogorov complexity.

Another popular approach to explaining generalisation is based around the idea of flat minima Keskar et al. (2016); Wu et al. (2017). In Hochreiter & Schmidhuber (1997), Hochreiter and Schmidhuber argue that flatness could be linked to generalization via the MDL principle. Several experiments also suggest that flatness correlates with generalization. However, it has also been pointed out that flatness is not enough to understand generalization, as sharp minima can also generalize Dinh et al. (2017). We show in Section 2 in the main text that simple functions have much larger regions of parameter space producing them, so that they likely give rise to flat minima, even though the same function might also be produced by other sharp regions of parameter space.

Other papers discussing properties of the parameter-function map in neural networks include Montufar et al. Montufar et al. (2014), who suggested that looking at the size of parameter space producing functions of certain complexity (measured by the number of linear regions) would be interesting, but left it for future work. In Poole et al. (2016), Poole et al. briefly look at the sensitivity to small perturbations of the parameter-function map. In spite of these previous works, there is clearly still much scope to study the properties of the parameter-function map for neural networks.

Other work applying PAC-Bayes theory to deep neural networks include (Dziugaite & Roy (2017; 2018); Neyshabur et al. (2017a;b)). The work of Dziugaite and Roy is noteworthy in that it also computes non-vacuous bounds. However, their networks differ from the ones used in practice in that they are either stochastic (Dziugaite & Roy (2017)) or trained using a non-standard variant of SGD (Dziugaite & Roy (2018)). Admittedly, they only do this because current analysis cannot be rigorously extended to SGD, and further work on that end could make their bounds applicable to realistic networks.

One crucial difference with previous PAC-Bayes work is that we consider the prior *over functions*, rather than the prior over parameters. We think this is one reason why our bounds are tighter than other PAC-Bayes bounds. The many-to-one nature of the parameter-function map in models with redundancy means that: low KL divergence between prior and posterior in parameter space implies low KL divergence between corresponding prior and posterior in function space; conversely, one can have low KL divergence in function space with a large KL divergence in parameter space. Using the PAC-Bayes bound from McAllester (1999b), this implies that bounds using distributions in function space can be tighter. As we explain in Section 5, we actually use the less-commonly used bound from McAllester (1998), which assumes the posterior is the Bayesian posterior, which we argue is a reasonable approximation. This posterior is known to give the tightest PAC-Bayes bounds for a particular prior (McAllester (1999b)).

Finally, our work follows the growing line of work exploring random neural networks Schoenholz et al. (2017a); Giryes et al. (2016); Poole et al. (2016); Schoenholz et al. (2017b), as a way to understand fundamental properties of neural networks, robust to other choices like initialization, objective function, and training algorithm.

