# OpenReview forum: "Deep learning generalizes because the parameter-function map is biased towards simple functions"
_ICLR.cc/2019/Conference_

### Official Review · AnonReviewer3 · 2018-10-30
**not surprising**

**Rating:** 4
**Confidence:** 4

**Review:**


The authors make a case that deep networks are biased
toward fitting data with simple functions.

The start by examining the priors on classifiers obtained by sampling
the weights of a neural network according to different distributions.  They do this
in two ways.  First, they examine properties of the distribution
on binary-valued functions on seven boolean inputs obtained by
sampling the weights of a small neural network.  They also empirically compare
the labelings obtained by sampling the weights of a network with
labelings obtained from a Gaussian process model arising from earlier
work.

Next, they analyze the complexity of the functions produced, using
different measures of the complexity of boolean functions.  A
favorite of theirs is something that they call Lempel-Ziv complexity,
which is measured by choosing an arbitrarily ordering of the
domain, writing the outputs of the function in that ordering,
and looking at how well the Lempel-Ziv algorithm compresses this
sequence.  I am not convinced that this is the most meaningful
and fundamental measure of the complexity of functions.
(In the supplementary material, they examine some others.
They show plots relating the different measures in the body
of the paper.  None of the measures is specified in detail in the
body of the paper. They provide plots relating these complexity
measures, but they don't demonstrate a very close connection.)

The authors then evaluate the generalization bound obtained by
applying a PAC Bayes bound, together with the assumption that
the training process produces weights sampled from the distribution
obtained by conditioning weights chosen according to the random
initialization on the event that they fit they fit the training
data perfectly.  They do this for small networks and simple datasets.
They bounds are loose, but not vacuous, and follow the same order
of difficulty on a handful of datasets as the true generalization
error.

In all of their experiments, they stop training when the training
accuracy reaches 100%, where papers like https://arxiv.org/pdf/1706.08947.pdf
have found that continuing training past this point further improves test
accuracy.  The experiments all use architectures that are
quite dissimilar to what is commonly used in practice, and
achieve much worse accuracy, so that a reader is concerned
that the results differ qualitatively in other respects.

I do not find it surprising that randomly sampling parameters
of deep networks leads to simple functions.

Papers like the Soudry, et al paper cited in this submission are
inconsistent with the assumption in the paper that SGD samples
parameters uniformly.

It is not clear to me how many hidden layers were used for the
results in Table 1 (is it four?).

I did find it interesting to see exactly how concentrated the
distribution of functions obtained in their 7-input experiment
was, and also found results on the agreement of the Gaussian process
models with the randomly sampled weight interesting, as far as they
went.  Overall, I am not sure that this paper provided enough
fundamental new insight to be published in ICLR.

---

> ### Author Response · Authors · 2018-11-28
> **Answer to reviewer (part 1)**
>
> We thank Reviewer 3 for the constructive comments and feedback.
>
> __not surprising?__
> The reviewer’s title says “not surprising” and in the text they write “I do not find it surprising that randomly sampling parameters of deep networks leads to simple functions.”
>
> We are happy that the reviewer does not find this surprising.  Be that as it may, to our knowledge we are the first to directly measure the parameter-function map for a DNN, showing simplicity bias over many orders of magnitude.  We demonstrate that the parameter-function map obeys the conditions that allow the simplicity bias bound to hold, which for biased maps, gives an exponential drop in probability with a linear increase in descriptional complexity.  (Note that it is not hard to see that many other machine learning methods don’t satisfy these necessary conditions, and therefore overfit when there are more parameters than data). We then show that this simplicity bias provides the implicit regularization that explains the remarkable generalization properties of  highly overparameterized DNNs.  In parallel, we provide a novel PAC-Bayes analysis based on the parameter-function map that generates the **first non-vacuous generalization bounds for DNNs that correctly scale with varying generalization performance for MNIST, fashion MNIST and CIFAR10**.  These bounds would not work unless the parameter-function map is extremely biased, an effect we capture in our version of PAC-Bayes for DNNs.   Regardless of whether or not the referee finds all this surprising, we believe that these results are significant, and have not been published in the literature on DNNs.
>
> __Complexity measures__
> The reviewer complains about our use of Lempel-Ziv (LZ), and that we put other complexity measures into the Appendix.   In our original manuscript we write “Here we simply note that there is nothing fundamental about LZ. Other approximate complexity measures that capture essential aspects of Kolmogorov complexity also show similar correlations (see Appendix E.4).”
> In the current manuscript we have expanded this sentence slighlty, but also note that the question of complexity measures has been further discussed in the Dingle et al (2018) paper we cite above.
>
> In more detail, as we write in the Appendix E.1 in the original manuscript “[the ordering] may affect the LZ complexity, although for simple input orderings, it will typically have a negligible effect.”,  Therefore, the ordering of the domain is not totally arbitrary. It must be a Kolmogorov-simple ordering to ensure that the complexity of the resulting bit string is close to the complexity of the function.
>
> Furthermore, we don’t claim that LZ is fundamental, or necessarily the best choice. The motivation behind using LZ is that it is commonly used to approximate Kolmogorov complexity, and it seemed to be the one that correlated best with the probability of Boolean functions, for the small fully connected network, although other measures (such as Boolean complexity) also do well.
>
> Regarding meaningfulness, we think some of the measures offered in the Appendix are perhaps more meaningful (or at least, interpretable) as they are truly only dependent on the function, and not domain ordering.
>
> At any rate, while It is true that the literature on complexity measures is vast, and much more could be said about them, for the basic argument we are making in the paper we  believe that the measures we use are sufficient.

---

> > ### Author Response · Authors · 2018-11-28
> > **Answer to reviewer (part 2)**
> >
> > __PAC-Bayes bounds__
> >
> > The reviewer says that “They bounds are loose, but not vacuous” -- To our knowledge, these are the first non-vacuous bounds for DNNs that follow the same trends as the generalization error.  We think this is a pretty big deal.  Most other bounds that follow trends are typically orders of magnitude larger than 1.  So we wouldn’t call this loose compared to the state of the field.
> >
> > __Early stopping__
> >
> > The reviewer says
> > “In all of their experiments, they stop training when the training accuracy reaches 100%, where papers like https://arxiv.org/pdf/1706.08947.pdf have found that continuing training past this point further improves test Accuracy.”
> >
> > While the paper cited is interesting, it mainly argues that certain bounds become better when training beyond zero training error, they don’t show that this holds for the true generalization error.  Moreover, the bounds they use are vacuous (>>1).  However there are papers that do directly discuss the generalization gain of longer training including https://arxiv.org/abs/1710.10345 and https://arxiv.org/pdf/1705.08741.pdf . The first paper only concerns itself with full batch gradient descent, not SGD. In both cases, the benefit of longer training is only a few percent improvement in generalization error.  There are many similar techniques that add a few percent to the generalization performance.  As explained in other responses above, we are not primarily writing about these small gains.
> >
> > __Realistic architectures__
> > The reviewer complains that
> > “ The experiments all use architectures that are quite dissimilar to what is commonly used in practice, and achieve much worse accuracy, so that a reader is concerned that the results differ qualitatively in other respects.”
> >
> > We disagree.  We use FC and CNN networks that are  similar to those used in practice.  It is true that the CNNs we use don’t have max-pooling, which is probably the main reason why their performance on CIFAR10 is less than state of the art.   We plan to extend our analysis to networks with pooling in future work. Moreover, for MNIST the performance is much closer to state of the art.  Of course we could push our results closer to the state of the art, but we don’t think this is necessary to make the main points of our paper.
> >
> > __SGD (and Soudry, et al.)__
> >
> > We disagree that Soudry et al. is inconsistent with our work.   See for example our discussion of SGD in the responses to referee 1 and in our general response. Generally, results suggesting that “optimization algorithms” are important in papers like that of Soudry, et al. are consistent with our work. When studying properties of any optimization algorithm like SGD, the parameter-function map plays a role. The better way to look at this is not as a mutually exclusive alternative, but as a new perspective that could shed light on old and new results. The perspective being that understanding properties of the parameter-function map can explain the observed behavior of a wide class of neural networks training algorithms.
> >   More specifically here, Soudry et al. look at full gradient descent, rather than stochastic gradient descent; it is not yet clear if the results would carry through to SGD. T
> >
> > __Others__
> >
> > The networks used for Table 1 are the same as in Figure 2, so the CNN has 4 layers, and the FC has 1 layer, we updated the caption to reflect this.

---

> > > ### Comment · AnonReviewer3 · 2018-12-01
> > > **read and considered**
> > >
> > > I have read and considered the author's response, and do not wish to change my rating of the paper.

---

### Official Review · AnonReviewer1 · 2018-11-02
**Interesting perspective but most relevant experiments are on very tiny networks**

**Rating:** 5
**Confidence:** 3

**Review:**

This paper propose an interesting perspective to explain the generalization behaviors of large over-parameterized neural networks by saying that the parameter-function map in neural networks are biased towards "simple" functions, and through a PAC-Bayes argument, the generalization behavior will be good if the target concept is also "simple". I like the perspective of view that combines the "complexity" of both the algorithm bias and the target concept in the view of generalization. However, the implementation and presentation of the paper could be improved.

First of all, the paper is a bit difficult to follow as some important information is either missing or only available in the appendix. For example, in Section 2, to measure the properties of the parameter-function mapping, a simple boolean neural network is explored. However, it is not clear how the sampling procedure is carried out. There is also a 'training set of 64 examples', and it not obvious to the reader how this training set is used in this sample of neural network parameters.

Following that, the paper uses Gaussian Process and Expectation-Propagation to approximately compute P(U). But the description is brief and vague (to non-expert in GP or EP). As one of the main contribution stated in the introduction, it would be better if more details are included.

Moreover, the generalization bound is derived with the assumption that the learning algorithm uniformly sample from the set of all hypothesis that is consistent with a given training set. It is unlikely that this is what SGD is doing. But explicit experiments to verify how close is the real-world behavior to the hypothetical behavior would be helpful.

The experiment in section 6 that verify the 'complexity' of 'high-probability' functions in the given prior is very interesting. It would be good if some kind of measurements more directly on the real world tasks could be done, which will better support the argument made in the paper.

---

> ### Author Response · Authors · 2018-11-28
> **Answer to reviewer**
>
> We thank Reviewer 1 for the constructive comments and feedback.
>
> __Tiny networks__
> The reviewer uses as a title “Interesting perspective but most relevant experiments are on very tiny networks”  -- While we do use a small model network for our direct sampling, we perform a significant amount of work on more standard architectures and datasets, including  4 hidden layer  CNNs  with 200 filters per layer for all databases, and a FC network that has 1 hidden layer, with 784 neurons for MNIST and fashion MNIST, and 1024 neurons for CIFAR10.    (We also used FC networks with more layers, but there results are similar, with only a small improvement in generalization).  While they are not the latest state of the art, we don’t think that these DNNs are tiny.
>
> __Clarity of exposition__
>
> We agree that we could have been clearer in what we are trying to achieve.  To this end, we have expanded the introduction, and throughout the paper tried to make our arguments more clear (see also our general response above).  In response to the reviewer, we have in particular improved the exposition in the sections which were a bit difficult to follow. In Sections 2 and 3, we explained the experiment and sampling procedure in more detail. In Section 2 we also defined the parameter-function map more clearly as per Reviewer 2’s advice. The mention of a “training set of 64 examples” was a typo, as the experiment in Section 2 did not involve any training.
>
> In response to the referee, we have expanded the description of the Gaussian processes (GPs) and the Expectation-Propagation in section 4.1 to help people unfamiliar with the topic. Nevertheless the link between DNNS and GPs  is a vast topic, going back to the famous 1995 work by Radfod Neal.  See also the pioneering recent papers we cite as [(Lee et al. (2017); Matthews et al. (2018); Garriga-Alonso et al. (2018); Novak et al. (2018))]. But  we hope that what we write, is sufficient for a non-expert to catch a flavour of the method.   Some more detail on GPs can be found in Appendix C, and of course m We are planning a longer publication explaining in much more detail how all this works for PAC-Bayes.
>
> __SGD__
>
> Here we quote the full paragraph on SGD because it raises an important issue that merits a longer response.  The  reviewer writes
> “Moreover, the generalization bound is derived with the assumption that the learning algorithm uniformly sample from the set of all hypothesis that is consistent with a given training set. It is unlikely that this is what SGD is doing. But explicit experiments to verify how close is the real-world behavior to the hypothetical behavior would be helpful.”
>
>
> __Our new experiments on SGD sampling__
>
> As also described above in our general response, in the new section 6, we performed experiments which test the behaviour of SGD in a more direct way than most previous approaches, at the expense of being constrained to very small input spaces (we use the neural network with 7 Boolean inputs and one Boolean output). We performed experiments directly comparing the probability of finding individual Boolean functions when training the neural network with two variants of SGD, versus using the Gaussian process corresponding to the neural network architecture (which approximates Bayesian sampling of the parameters under i.i.d. Gaussian prior). We find good agreement.
>
> __Complexity of real-world functions__
>
> The reviewer also asks for direct measurements of the complexity of real world functions.  This is indeed an interesting question.  While the simplicity bias bound means that large P(f) must mean low complexity, it is not so easy to calculate the complexity for  real world functions using most of the measures we consider in the Appendix.   We are currently working on this question using the critical sample ratio, which is the most scalable measure.  Preliminary results are encouraging, but they weren’t ready by the deadline to submit the manuscript..
> It’s not hard to imagine that function for in Figs 3 and 4 are more complex as we corrupt the data more.

---

> > ### Author Response · Authors · 2018-11-28
> > **Other papers on SGD that are relevant**
> >
> > __Other papers on SGD that are relevant__
> >
> > We also include in the introduction a brief description of a related argument by (Wu et al. 2017 https://arxiv.org/abs/1706.10239 ) who found that normal gradient descent (GD) and SGD gave  similar results for several architectures on MNIST.  In their paper they find a wide range of generalization performance that correlates very well with a measure of local flatness.  They point out that flatness correlates with basin volume V (a concept from optimisation on landscapes) and since the volume varies a lot they argue that  both GD and SGD, at least to first order, find basin volumes V that are large, and so find similar generalization performance.  In our work we directly calculate the volume via P(f), which is proportional to the volume of parameter space that generates function f.  This volume is also correlated with the basin volume V, and we make a similar qualitative argument, namely that the very large bias in the parameter-function map  is likely to be the first order driver of what solutions SGD finds.
> >
> > (As an aside, we note that while the concepts of flatness are vulnerable for example to reparameterization, and moreover are local, P(f) is a global property)
> >
> > We are planning a longer paper on the complex question of how SGD samples parameters. It may nevertheless be helpful to include some further discussion of the literature here.
> >
> >  So far there have not been many works that directly study how SGD samples parameters.  Nevertheless, there are interesting indirect suggestions in the literature that are worth exploring.
> > For example, in a study comparing GPs and direct SGD on a CNN and an FCN, all trained on CIFAR10,  (see Figure 4 (b) of Nowak et al. (2018)  https://arxiv.org/pdf/1810.05148.pdf ) the authors show that the GPs and SGD are very close in generalization accuracy once enough channels are included in the CNN.  Since the Gaussian processes assume a Gaussian prior on parameters, and then weight according to the likelihood on the training data, they are  essentially sampling the parameter-function map in the same way that we are assuming is happening for the PAC-Bayes bounds.   Thus these results are highly suggestive of SGD also sampling parameters in  way that may approach direct uniform sampling of parameters.
> >  These results have been further discussed in Matthews et al (2018) https://arxiv.org/pdf/1804.11271.pdf  where on p3 the authors write “Lee et al.(2018) compare finite neural networks trained with stochastic gradient descent (SGD) to Gaussian processes instead. The latter comparison to SGD is suggestive that this optimization method mimics Bayesian inference – an idea that has been receiving increasing attention (Welling and Teh, 2011; Mandt et al., 2017; Smith and Le, 2018).”
> >
> > However, there are many subtleties, also shown in Nowak et al. (2018). See e.g. their tables 1 and 2.  Overall there is good agreement for the generalisation performance of GPs and  more standard CNNs and FCNs. However,  by careful hyperparameter tuning, including pooling, by having the SGD underfit the training data, or by combining high learning rates with ReLU non-linearities, they do obtain better results than for GPs for the CNNs.  These  results show that there is more to understand about how SGD samples parameters.  It would be interesting to see how much better the GPs work if pooling or underfitting training data is included.  It might be, for example, that since underfitting training data constrains the solutions less, that lower complexity functions are found.   In the PAC-Bayes language this leads to a larger P(U), which may sometimes compensate for the increase in training error.
> >
> > There are also other examples of where SGD differs from Bayesian inference. For example experiments in Mandt et al. (2017)  https://arxiv.org/abs/1704.04289 find that the SGD posterior differs from the Gibbs posterior. But in order to do this analysis, the authors needed to use very low dimensional spaces, which may be under-representative of high-dimensional parameter spaces of neural networks.
> >
> > Finally, we are well aware of a significant stream of the literature which conjectures that certain properties SGD are the dominant source of generalisation in DNNs.  While SGD is unquestionably very important for optimisation, the fact that other optimisation methods lead to similar generalisation (see e.g. some of the discussion in our paper) suggest that SGD is not the main source of generalization.
> >
> > In summary, while more work needs to be done,  we believe that we are justified in our conjecture that SGD samples the highly biased parameter-function map close enough to the way Bayesian sampling would sample functions that we can apply PAC-Bayes bounds to simple SGD.   If we are wrong, then there must be some interesting cancellation of errors, as our bounds work remarkably well.

---

### Official Review · AnonReviewer2 · 2018-11-02
**A fresh study to the generalization capabilities of (deep) neural networks, with the help of the PAC-Bayesian learning theory and empirically backed intuitions.**

**Rating:** 7
**Confidence:** 4

**Review:**

The paper brings a fresh study to the generalization capabilities of (deep) neural networks, with the help of an original use of PAC-Bayesian learning theory and some empirically backed intuitions.

Expressing the prior over the input-output function space generated by the neural network is very interesting. This provides an original analysis compared to the common PAC-Bayesian analysis of neural networks that express the prior over network parameters space. The theoretical study here appears simple (noteworthy, it is based one of the very first PAC-Bayesian theorems of McAllester that is not the most used nowadays), and the study is conducted mainly by empirical observation. Nevertheless, the experiments leading to these observations are cleverly designed, and I think it gives great insights and might open the way to other interesting studies.

Overall, the paper is enjoyable to read. I also appreciate the completeness of the supplementary material. I recommend the paper acceptance, but I would like the authors to consider the concerns I rise below:
- The paper title is a bit presumptuous. The paper presents a conjunction backed by empirical evidence on some not-so-deep neural networks. Even if I consider it as an important piece of work, it does not provide any definitive answer to the generalization puzzle.
- Many peer-reviewed publications are cited as arXiv preprints. Please carefully complete the bibliography. Some papers are referenced by the name, title and year only (Smith and Le 2018; Zhang et al, 2017)
- I recommend adding to the learning curves of Figures 2 and 3 the loss on the training set.

Other minor comments and typos:
- Intro: Please define "parameter-function" map
- Page 4: Missing parentheses around Mand et al. (2017)
- SGD has not had time ==> SGD did not have time
- Please refers to the definition in the supplementary material/information the first time you mention Lempel-Ziv complexity.
- Please mention that SI stands for Supplementary Information

---

> ### Author Response · Authors · 2018-11-28
> **Answer to reviewer**
>
> We thank Reviewer 2 for the constructive comments and feedback.
>
> We have submitted a new draft were we address  concerns raised, and sharpen some of the main points we make.   In particular, we have clarified what we are trying to explain in terms of the generalization puzzle.  We are trying to explain the big picture of why overparametrized DNNs generalize at all, and have tried to clarify in the text that we are not trying to explain for example why SGD or dropout or other similar techniques improve further on generalization.    We’re still happy to add qualifiers to the title if the reviewer wants us to.
>
> We have fixed the bibliography, correctly citing peer-reviewed publications, and completing those with incomplete information.
>
> The classification error on the training set is 0 in all the experiments in Figures 2 and 3. We have updated the captions to make this clear in the figures themselves. If instead ‘loss’  referred to the cross-entropy loss, that will of course not be exactly zero, but our discussion is centered on classification error, and so we think adding that would be distract from the point of the figures.
>
> We have added a definition of parameter-function map in Section 2, and addressed all the other minor typos and comments. We also changed to “Supplementary information” to Appendices.

---

### Author Response · Authors · 2018-11-28
**Overview of changes**

We thank the reviewers for their constructive comments and feedback, which stimulated us to improve our paper.  Here we list the main changes:

1) We have rewritten the abstract and significantly expanded the introduction to make it clearer the question we are trying to answer:   **Why do highly over-parameterised deep neural network (DNN) generalize at all, given that the expectation from classical learning theory is that such highly expressive models should strongly overfit?**  Our answer to this generalization puzzle is that DNNs exhibit a strong intrinsic bias towards simple functions that provides the main source of implicit bias needed to explain the puzzle of generalization.   There are many empirical results showing, for example, that using dropout or using SGD instead of gradient descent (GD), or using early stopping etc...  leads to improvements in generalization. While important for practical applications, these improvements are generally relatively small, and so don’t answer the big question that we are trying to address.

2) In section 2, we have added a clearer definition of the parameter-function map, which we argue provides a novel and fruitful lens through which to analyze generalization in DNNs.

3) In section 3, we  improved our description of how the AIT-inspired  simplicity bias phenomenology from (K. Dingle, C. Q. Camargo and A. A. Louis, Nature Comm. 9, 761 (2018)) applies to the parameter-function map of DNNs.  In particular  simplicity-bias predicts that functions f with a relatively high probability P(f) to obtain upon random sampling of parameters will have a relatively low descriptional complexity.  **Our key argument is that such easy-to-find functions will also generalize well**.    We demonstrate that this works explicitly for our smaller model network with two hidden layers of 40 neurons each (that nonetheless can express on the order of 10^(38) functions)..  Since the AIT based arguments for simplicity bias are very general, they should apply for larger DNNs as well where direct sampling is out of the question.

4) In section 4 we have  expanded our description of how we use Gaussian processes (GPs)  to calculate the probability P(U) of the data, which plays a key role in our PAC-Bayes bounds, which, in turn provide an independent argument for why highly biased parameter-function maps lead to good generalization performance.

5) We made no major changes to the results section, but do note that our CNN sizes are 4 hidden layers with 200 filters per layer for all databases, while the FC network had 1 hidden layer, with 784  neurons for MNIST and fashion MNIST, and 1024 neurons for CIFAR10.    (We also used FC networks with more layers, but  results are similar, with only a small improvement in generalization, so we didn’t show these).

6) We have added a new section 6 where we directly compare the probability of finding individual functions using SGD with an estimate using GPs.  While the agreement is not exact, which could be due to errors in our GP calculation (see Fig 6 in Appendix B), or due to deviations of SGD from Bayesian sampling, overall the trend is encouraging. We note that the probabilities range over many orders of magnitude.   In PAC-Bayes, the bias enters the PAC-Bayes via a log, so we only need SGD to be similar to Bayesian sampling on a log scale.  Thus we believe that the agreement we find here between SGD and the GP prior is good enough for PAC-Bayes to work.   One can also turn this argument around: The fact that we find, for the first time for DNNs, non-vacuous (< 1) bounds using PAC-Bayes is highly non-trivial, and provides indirect evidence that SGD is indeed sampling functions roughly consistently with the prior P(f).

7) We have slightly sharpened the conclusions, but not changed this section significantly.

8) We have moved a section on choice of hyperparameters for the GP from the main text to the Appendix C, as it was quite technical and not central to our main argument.  We also added Appendx D, which explains how we compare function probabilities for GPs to SGD, as well as Appendix I,  Bias and the curse of dimensionality, where we discuss why other machine learning methods that are not biased do suffer from overfitting, in contrast to DNNs.

---

### Meta-Review · Area_Chair1 · 2018-12-13
**An interesting addition to the deep learning theory literature**

**Confidence:** 4
**Recommendation:** Accept (Poster)

**Metareview:**

Dear authors,

There was some disagreement among reviewers on the significance of your results, in particular because of the limited experimental section.

Despite this issues, which is not minor, your work adds yet another piece of the generalization puzzle. However, I would encourage the authors to make sure they do not oversell their results, either in the title or in their text, for the final version.